# ALM-MTA: Front-Door Causal Multi-Touch Attribution Method for Creator-Ecosystem Optimization

**Yuguang Liu**[1][*] **Luyao Xia**[2][*] **Hu Liu**[1][*] **Zhangxi Yan**[1] **Jian Liang**[1] **Han Li**[1] **Kun Gai**[1]

[1]Kuaishou Technology [2]The University of Auckland

{liuyuguang,yanzhangxi,liangjian03,lihan08,yuyue06}@kuaishou.com
luyao.xia@auckland.ac.nz, hooglecrystal@126.com

## ABSTRACT

Consumption Drives Production (CDP) on social platforms aims to deliver interpretable incentive signals for creator ecosystem building and resource utilization improvement, which strongly relies on attributions. In large-scale and complex recommendation system, the absence of accurate labels together with unobserved confounding renders backdoor adjustments alone insufficient for reliable attribution. To address these problems, we propose Adversarial Learning Mediator based Multi Touch Attribution (ALM-MTA), an extensible causal framework that leverages front-door identification with an adversarially learned mediator: a proxy trained to distillate outcome information to strengthen causal pathway from treatment to outcome and eliminate shortcut leakage. Then, we introduce contrastive learning that conditions front door marginalization on high match consumption upload pairs for ensuring positivity in large treatment spaces. To assess causality from non RCT logs, we also incorporate a non personalized bucketed protocol, estimating grouped uplift and computing AUUC over treatment clusters. Finally, we evaluate ALM-MTA performance using a real-world recommendation system with 400 million DAU (daily active users) and 30 billion samples. ALM-MTA has increased DAU with 0.04% and 0.6% of the daily active creators, with unit exposure efficiency increased by 670%. On causal utility, ALM-MTA achieves higher grouped AUUC than the SOTA in every propensity bucket, with a maximum gain of 0.070. In terms of accuracy, ALM-MTA improves upload AUC by 40% compared to SOTA. These results demonstrate that front-door deconfounding with adversarial mediator learning provides accurate, personalized and operationally efficient attribution for creator ecosystem optimization.

## 1 INTRODUCTION

In platform-mediated creator economies, content platforms have exhibited a stable 'consumption-drives-production' pattern: users are more likely to transition into creation (i.e., upload new content) after consuming sufficiently 'inspiring' content Yao et al. (2024); O'Toole (2023). Accordingly, understanding and quantifying the drivers of user-generated content enables platforms to present content that inspires creation, thereby cultivating a vibrant creator ecosystem and optimizing resource allocation O'Toole (2023); Bhargava (2022). Within recommender systems, the task is naturally formalized as a multi-touch attribution (MTA) problem, in which multiple prior touchpoints in a user's consumption sequence may jointly lead to the eventual conversion (an upload) Shender et al. (2020); Yao et al. (2022). Accurate causal attribution is not only essential for interpretability, but it also directly impacts on incentive design, cold-start handling, and re-ranking optimization Liu et al. (2025a). However, in large-scale, complex recommender systems, multi-touch attribution is jointly constrained by (i) the absence of explicit ground truth, (ii) pervasive multi-source confounding, and (iii) the large-scale candidate-touchpoint space, rendering reliable and scalable causal attribution a central challenge for system design and creator-ecosystem governance Luo et al. (2024).

Within these constraints, industrial practice typically bifurcates attribution into strict rule based methods and semantic/path similarity methods. The former delivers high precision yet very limited coverage, whereas the latter broadens coverage but lacks causal identifiability and often con-

---

[*]Equal contribution

flates correlation with cause Gao et al. (2024). Both methods fail to estimate the counterfactual effect on the upload outcome of removing a specific video within the video consumption sequence, while preserving the rest of the trajectory (see Fig.1). At the individual level, estimating a personalized causal effect requires quantifying, within a user's sequential consumption context, each touchpoint's marginal uplift in that user's upload probability producing an intra-individual ranking by causal strength Wang et al. (2025); Tang (2024). Since rule- or semantic-based methods typically rely on cohort-level averaging and similarity scores, ignoring heterogeneous treatment effects (HTE), sequence effects, and interactions among touchpoints, they are ultimately difficult to provide personalized effect estimates at the user level Lee et al. (2025); Zhan et al. (2024). Consequently, conventional methods cannot simultaneously achieve precision, coverage, and pwrapfigureersonalization.

To address these challenges, we present a novel counterfactual causal multi-touch attribution framework that defines a touchpoint's uplift as the drop in predicted upload probability when that touchpoint is deleted from the video consumption sequence. Given unobserved confounding in complex systems, backdoor adjustment alone cannot ensure causal identifiability. We therefore invoke front-door identification by introducing a latent mediator

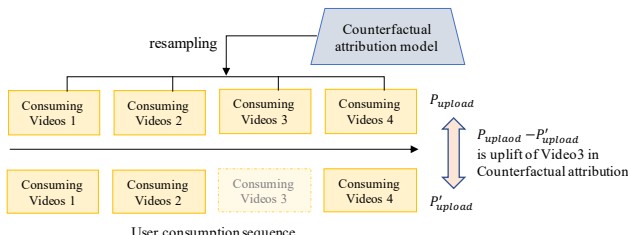

Figure 1: Counterfactual attribution by touchpoint deletion in the video consumption sequence.

that transmits the causal pathway from consumption to upload. This mediator can be observed through an adversarial, label-agnostic proxy $Y'$ that does not reveal the outcome $Y$. We ensure scalability and overlap in high-cardinality treatment spaces through contrastive learning and calibrated reweighting, and train for stability under personalization and platform dynamics. The result is individualized and reproducible uplift estimates that unify attribution across diverse business tasks.

In summary, this paper presents the following key contributions:

- **A novel causal MTA framework** that addresses unobserved system-level confounders in recommendation ecosystems, overcoming a core limitation of prior attribution methods.

- **A unified strategy for eliminating latent confounding** that combines front-door identification, adversarial proxy mediation, and propensity reweighting. This design ensures causal identifiability in the presence of unobserved confounders and prevents outcome leakage by making the mediator observable.

- **Scalable overlap-preserving operationalization.** A deployment-ready learning and evaluation method that maintains positivity and consistency in sparse, larger treatment.

- **Strong empirical gains at real-world large-scale recommendation system.** ALM-MTA (i) expands attributable coverage by 1.5× over the prior SOTA, (ii) increases daily active users (+0.04%) and daily active creators (+0.6%), (iii) improves unit exposure efficiency by 670%, (iv) achieves higher grouped AUUC across all propensity buckets (maximum gain +0.070), and (v) attains an upload AUC of 0.907 (+40% over SOTA).

## 2 RELATED WORK

**Data-Driven Multi-Touch Attribution.** The central task of MTA is to allocate credit for a conversion across a sequence of user touchpoints. Early data-driven methods moved beyond simple heuristics (e.g., last-touch) to employ interpretable statistical models. Foundational work includes using logistic regression and survival analysis to model conversion probability based on touchpoint history Shao & Li (2011); Ren et al. (2018). To better capture the sequential and temporal nature of user journeys, subsequent research adopted path-level models. Markov chains, which model the probability of transitioning between channels, became a popular approach for quantifying a channel's marginal contribution through its removal effect Anderl et al. (2014). This line of work was later extended with more powerful sequence models like Recurrent Neural Networks (RNNs) to learn more complex user path representations Arava et al. (2018); Ji & Wang (2017). Concurrently, game-theoretic approaches, particularly the Shapley value, gained widespread adoption due to their

axiomatic properties of fairness and additivity Zhao et al. (2018). However, the computational complexity of the exact Shapley value has spurred research into efficient approximation methods Jia et al. (2019); Kumar et al. (2020). While these methods offer valuable insights, they are predominantly correlational and often rely on the strong, implicit assumption that user touchpoints are exogenous, failing to adequately address the confounding bias inherent in real-world systems.

**Causal Inference in Attribution and Recommendation.** More recent work has rigorously framed MTA as a causal effect estimation problem, aiming to compute the uplift or Average Treatment Effect (ATE) of a touchpoint on a user's conversion probability. A dominant approach for this is the backdoor adjustment, rooted in the potential outcome framework Chen (2024). This involves leveraging methods like Inverse Propensity Weighting (IPW) and Doubly Robust (DR) estimators to control for observed confounders Rosenbaum & Rubin (1983); Dudík et al. (2011). These techniques are foundational to modern causal MTA and efforts to debias recommender systems from selection and exposure biases Bottou et al. (2013); Schnabel et al. (2016). However, their validity hinges on the strong ignorability (or unconfoundedness) assumption—that all common causes of treatment and outcome have been measured. This assumption is rarely tenable in complex ecosystems like recommender systems, where latent factors such as user intent, social influence (peer effects), and the platform's own strategic biases act as unobserved confounders Yang et al. (2020); Molina et al. (2022). Our work directly confronts this critical limitation.

**Front-Door and Proxy-Based Methods for Unobserved Confounding.** When strong ignorability is violated, alternative identification strategies are required. The front-door criterion, introduced by Neuberg (2003) provides another powerful theoretical tool for identifying causal effects in the presence of unobserved confounding, provided a valid mediating variable can be identified. A mediator is a variable that intercepts the causal pathway from treatment to outcome entirely. While theoretically elegant, the practical application of the front-door criterion has been limited, primarily due to the stringent requirement of finding and perfectly measuring a mediator that satisfies the necessary conditional independence properties Xu et al. (2023b). Recent efforts have explored using deep learning models to learn mediator representations, but often still assume the mediator is fully observable Luo et al. (2024); Xu et al. (2023c). To overcome the challenge of unobservable variables, researchers have turned to proxy-based and adversarial learning techniques. Proxy variables correlated with latent confounders have been investigated to partially mitigate confounding bias Miao et al. (2018a); Tchetgen Tchetgen et al. (2024a). In parallel, adversarial learning has been adapted for causal inference to learn balanced representations that are invariant to treatment assignment, thereby mitigating confounding bias Ganin et al. (2016); Long et al. (2017); Johansson et al. (2016); Shalit et al. (2017). However, they have not been systematically integrated with the front-door criterion for MTA. Our work present a framework that leverages an adversarial proxy to render the latent mediator observable while preventing outcome leakage, thereby operationalizing the front-door criterion for MTA. In contrast to prior proxy-based methods that primarily approximate unobserved confounders to mitigate bias, our design focuses on enabling mediator observation as the key to front-door identifiability. Recent debiasing methods address latent confounding through multi-cause modeling Huang et al. (2025) or latent causal structure learning Xu et al. (2025), which rely on backdoor-style adjustment. Moreover, including substitute confounders Xu et al. (2023a) method, these approaches do not address front-door identification with latent mediators, outcome leakage from proxies, or positivity in high-cardinality treatments. ALM-MTA fills these gaps through adversarial proxy mediation and contrastive overlap control.

## 3 TASK AND PROBLEM FORMULATION

**Task.** Given a user's recent consumption sequence and contextual features, we estimate the individualized causal uplift of each touchpoint (a consumed item) on the user's subsequent upload (creation) behavior and output this uplift as multi touch attribution for interpretation and downstream ranking. Specifically, let the user level observation be $(X, T, Y)$: $X$ denotes user and environmental context (including observable platform covariates), $T = \langle \tau_1, \ldots, \tau_L \rangle$ is time-ordered sequence of consumed touchpoints, and $Y \in \{0, 1\}$ indicates whether the user uploads within consumption sequence horizon. The uplift of consumed touchpoint is defined as the counterfactual removal drop in upload probability. See Appendix §A.1 for a detailed discussion of the front-door assumptions and the proxy mediator.

$$\Delta(\tau_j | X, T) = P(Y = 1 | do(T), X) - P(Y = 1 | do(T \backslash \tau_j), X). \tag{1}$$

In practice, the model outputs the approximate difference between these two counterfactual probabilities as the attribution strength, producing "multi-point" attribution labels that construct actionable touchpoint-level causal attribution targets in non-RCT logs without explicit labels.

**Causal Graph Structure and Variables.** Large-scale recommendation involves system- and sequence-level confounding which is hard to enumerate. To obtain identifiable individual effects, we adopt a graph $(X, W, T, M, Y)$ with unobserved confounders $W$, mediator $M$, and outcome $Y$(see Fig.2). $X$ is observable confounding (e.g., static or dynamic attributes of users), $W$ is unobservable confounding (e.g., strategies of recommendation systems), $M$ is mediator (e.g., the path to trigger the upload action after a user views a video in CDP), $Y'$ is the proxy (e.g., compare the upload with the videos user has viewed and determine the motivated path). In the Causal Graph, the unobserved confounder $W$ and the observed covariates $X$ can both influence $T$ and $Y$, and the causal path from touchpoints to the outcome must pass through $T \to M \to Y$. Since large-scale recommendation system involves hard-to-enumerate system- and sequence-level confounding, backdoor adjustment alone cannot ensure identifiability.

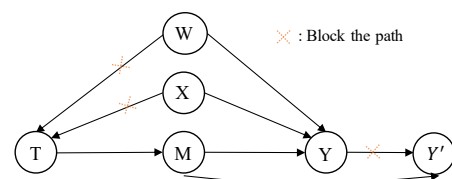

Figure 2: Causal graph with latent confounding and adversarially observed mediator. $X$ denotes observed confounding. $W$ is unobserved potential confounding. $T$ means treatment. $Y$ is the result or output variable. $Y'$ is observations of result $Y$. $M$ represents Mediator, which is transmission path between T and $Y$.

**Front-Door Identification.** To identify $P(Y = 1|do(T), X)$ and $\Delta(\tau_j|X, T)$ thereby under the observed distribution, we invoke the front-door criterion: (i) Path containment: all directed paths from $T$ to $Y$ pass through $M$; (ii) No backdoor for $M \to Y$: unobserved confounding does not affect $M$ (or is blocked by $X$); (iii) Block $T \to M$ backdoors: by marginalizing on $T$ blocks backdoor paths from $T$ to $M$. Under these conditions, the standard front-door formula gives:

$$E(Y|do(T = t)) = \Sigma_m E(Y|do(M = m)) * P(M = m|do(T = t)), \tag{2}$$

from which touchpoint-level uplift is defined and estimated.

**Overlap under large-cardinality treatments.** The meaning of attribution overlap is that for any unit, the probabilities of receiving and not receiving the treatment are strictly positive. When marginalizing over $T$, the denominator of the conditional probability must be non-zero. However, in the bandwagon business scenario, the treatment space is extremely large, making it difficult to satisfy the overlap condition when marginalizing $T$.

**Objective (individualized attribution).** We estimate a function $g_\theta(X, T)$ that approximates the interventional response $P(Y = 1|do(T), X)$ via front-door identification, integrating the adversarial proxy-mediator and contrastive filtering into the estimation pipeline. Touchpoint-level attribution for $\tau_j \in T$ is defined as:

$$\hat{\Delta}(\tau_j|X, Y) = g_\theta(X, T) - g_\theta(X, T\{\tau_j\}). \tag{3}$$

## 4 MODEL: ALM-MTA FRAMEWORK

ALM-MTA (Fig.3) can be viewed as a practical front-door generalization of causal multi-touch attribution to large-scale recommendation, taking a user context $X$ and a time-ordered treatment sequence $T$ to produce individualized, touchpoint-level uplift labels. Operationally, the model first reweights observational logs with inverse propensity scores (IPW) to mitigate back-door bias from $X \to T$. A shared neural backbone then encodes the sequence and feeds two coordinated heads. The first is a lightweight ITE head that aggregates touchpoint embeddings with learned weights to parameterize the logit for $P(Y|X, T)$. Uplift is obtained by comparing this logit with the one re-estimated after removing a touchpoint. The second is a mediator-observation head supervised by a proxy $Y'$. Through adversarial learning, this head is trained to capture the $T \to M \to Y$ pathway while suppressing shortcut leakage $Y' \to Y$ so that the mediator signal remains usable for front-door identification.

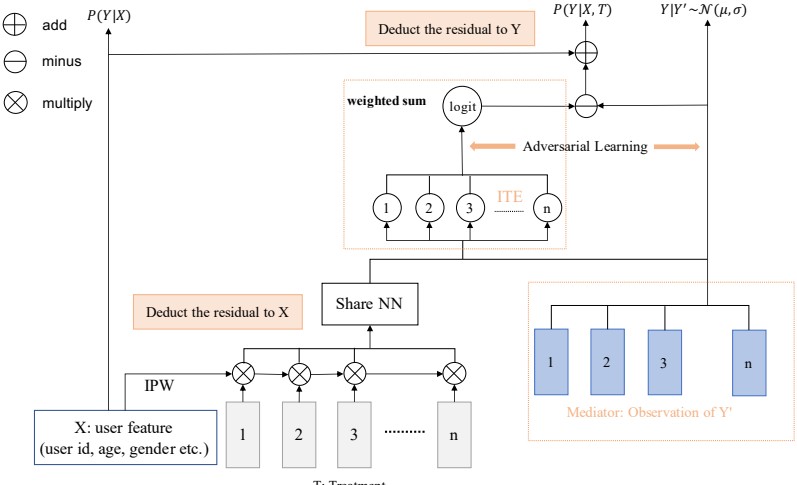

Figure 3: The ALM-MTA architecture. User features and treatment sequences are reweighted via IPW and encoded by a shared backbone. A weighted ITE head estimates per-touchpoint uplift, while an adversarial proxy branch renders the latent mediator observable without leaking the outcome. Residual corrections refine signals, jointly enabling front-door identification and stable counterfactual attribution at scale.

### 4.1 THE STRATEGIES FOR ELIMINATING LATENT CONFOUNDING

A major challenge in multi-touch attribution in observational logs is the presence of latent confounders that simultaneously affect the consumed touchpoints $T$ and the upload result $Y$. Without proper treatment, these confounders lead to biased attribution estimates. We adopt three strategies, including front-door identification, adversarial proxy mediation, and propensity reweighting, to eliminate such confounding and guarantee causal identifiability.

**Eliminating latent confounding via the front-door criterion.** As shown in Section 3, the causal graph in Figure 2 fully satisfies the three conditions of the front-door criterion. Therefore, we can apply the front-door criterion to eliminate unobserved confounders $W$. When conditioning on treatment $T$ and covariates $X$, the mediator $M$ is not confounded with the result $Y$. Formally,

$$E[Y|do(M = m)] = \Sigma_{t',x}E[Y|M = m, T = t', X = x] * P(T = t', X = x). \tag{4}$$

Given $T$ and $X$, all back-door paths from $M$ to $Y$ are blocked, so that

$$E[Y|do(M = m), T = t', X = x] = E[Y|M = m, T = t', X = x], \tag{5}$$

$$\begin{aligned}E[Y|do(M = m)] &= \Sigma_{t',x}E[Y|M = m, T = t', X = x] * P(T, X|do(M = m)) \\ &= \Sigma_{t',x}E[Y|M = m, T = t', X = x] * P(T, X).\end{aligned} \tag{6}$$

Substituting this equality into the front-door formula yields the following.

$$\begin{aligned}E[Y|do(T = t)] &= \Sigma_{m,t',x}E[Y|M = m, T = t', X = x] \\ &\quad \times P(T = t', X = x) * P(M = m|T = t).\end{aligned} \tag{7}$$

This derivation shows that the causal effect of $T$ on $Y$ can be identified through the mediator $M$, even in the presence of unobserved confounders $W$.

**Mediator Identification via Adversarial Proxy.** We posit that the causal effect of $T$ on $Y$ is transmitted through a latent mediator $M$. However, $M$ is unobservable in practice. We therefore introduce a proxy variable $Y'$ that correlates with $M$. A mediator branch is supervised by $Y'$ to produce an embedding $\hat{M}$ To prevent shortcut leakage, we add an adversarial objectiveBai et al. (2021): a discriminator attempts to predict $Y$ from $\hat{M}$, while the mediator branch is optimized to suppress this predictability. As a result, $\hat{M}$ retains only the information necessary for the causal path $T \rightarrow M \rightarrow Y$, without directly revealing the outcome. Observing the proxy brings an additional benefit. By the monotonicity of conditional variance, enlarging the conditioning set from $\{X, T\}$ to $\{X, T, Y'\}$ satisfies:

$$Var(Y|X, Y) \geq Var(Y|X, T, Y'), \tag{8}$$

which shows that incorporating $Y'$ reduces the variance in the estimation of uplift and improves stability in high-cardinality treatments. This is particularly valuable in high-cardinality treatments, where data per touchpoint is sparse.

**Unconfoundedness under the observational distribution.** Data-driven modeling must rely on historical exposure logs; because exposures are jointly determined by user preferences and platform policies, the interventional path $do(T)$ deviates significantly from the empirical log $P_{\text{obs}}(X, T, M, Y)$. Direct training on the raw data therefore learns correlation and is affected by confounding. To bridge this gap using observational data alone, we adopt a "front-door debiasing + inverse-propensity reweighting" scheme: assign each sample the weight $w(x, t) = 1/P_{\text{obs}}(T = t \mid X = x)$. Under positivity and a correctly specified propensity model, the reweighted distribution becomes "as if randomized" given $X$; further leveraging the front-door structure $T \to M \to Y$ (which, after conditioning on $T, X$, blocks all back-door paths from $M$ to $Y$), the upload probability can be decomposed into counterfactual gains at the touchpoint level that are estimable from observables (see Appendix §A.2 for derivation):

$$
\begin{aligned}
\text{upload} = \sum_t \text{uplift}_t &= \sum_t E[Y \mid do(T = 1)] - E[Y \mid do(T = 0)] \\
&= \sum_t \sum_{x,m} f(m, t, x) \frac{P_{\text{obs}}(x, m, T_{\text{origin}})}{P_{\text{obs}}(t \mid x)} = \sum_{\text{instance}} f(M, T, X) \, w(X, T).
\end{aligned} \tag{9}
$$

Where $f(M, T, X) = E[Y \mid M, T, X]$. It follows that combining front-door debiasing with IPW enables causal attribution to be identified directly from raw logs, guarantees unconfoundedness with respect to latent confounders, and establishes the scheme's theoretical implementability. The loss in upload probability is modeled as the uplift, which is assumed to be linearly aggregable into the overall upload.

## 4.2 Contrastive Learning for Overlap

For causal identification, every unit must have non-zero probability of receiving each treatment. In large-scale attribution, however, the treatment space is extremely high-cardinality, and many touchpoints occur rarely. This leads to violations of positivity, making marginalization over all $T$ unstable and high-variance. Therefore, we introduce a contrastive learning module that estimates the semantic 'match' between each consumed touchpoint $T$ and the proxy mediator $Y'$, as shown in Fig.4. Positive pairs are constructed from the actual proxy–touchpoint interactions that precede an upload, while negatives are drawn from unrelated touchpoints. The InfoNCE loss encourages embeddings of $(Y', T)$ pairs that are causally related to being closer, and unrelated pairs to be farther apart. The resulting score $\omega(\tau, Y')$ quantifies how strongly a touchpoint is aligned with the proxy outcome. During front-door estimation, we restrict marginalization to the high-match subset $\tau_{high} = \{\tau \in T | \omega(\tau, Y') top - K\}$, and reweight accordingly. This guarantees that the denominator of conditional probabilities remains positive, satisfying overlap and reducing estimator variance.

## 4.3 Stability-Oriented Attribution

Causal conclusions should not fluctuate simply because of incidental variations in training. When the same statistical procedure is applied, the resulting causal effects ought to remain invariant. In practice, models trained on large-scale personalized logs are highly sensitive that uplift estimates often drift across random seeds, data orderings, and minor perturbations, undermining the reproducibility of attribution. Our framework explicitly enforces stability by designing the learning architecture and training process to resist such sources of randomness. Instead of allowing the network to absorb all correlations present in the data, we deliberately simplify the feature interactions so that only robust signals are retained. This

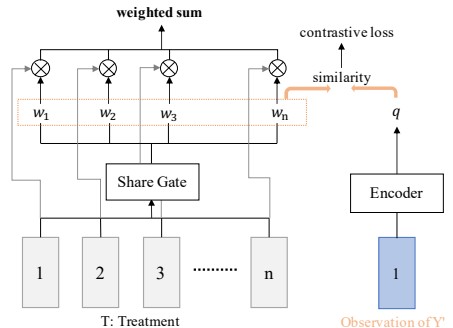

Figure 4: Contrastive overlap control for front-door estimation.

structural parsimony is further reinforced by $L2$ regularization, which prevents the model from overfitting spurious fluctuations.

## 5 EXPERIMENTS

### 5.1 BASELINE MODEL

We benchmark three families of methods on the Criteo conversion-prediction task: statistical learning (SL), deep learning (DL), and causal learning (CL). Logistic Regression (LR) serves as the classical statistical learning baseline. For deep learning, we include deepMTA Neuberg (2003) as a representative attribution model. We evaluate three causal learning baselines: DML Chernozhukov et al. (2018), DESCN Zhong et al. (2022), and causalMTA Liu et al. (2025a). All baselines are trained on the same data and features, using identical splits and evaluation protocols for fair comparison.

### 5.2 EXPERIMENT SETUP

The model comprises a counterfactual attribution backbone adversarial- and contrastive-learning branches. The backbone estimates per-touchpoint uplift under deletion/perturbation on list-wise sequences and auxiliary branches perform proxy mediation and cross-modal alignment between consumed items and produced content. The per-forward complexity is 1.31B FLOPs in total: 0.92B FLOPs for the counterfactual attribution backbone and 0.39B FLOPs for adversarial and contrastive components. Path-level attribution signals are subsequently converted into point-wise targets for scalable distributed training. Simultaneously, we use the Direct Routing GradientLiu et al. (2025b) method to reduce the impact of tradeoffs between targets. Further implementation details and the full training recipe are provided in Appendix A.3.

### 5.3 DATA

Our experiments are conducted on large-scale user consumption–production logs. To simulate the streaming nature of online systems, we adopt a streaming training method with both positive (upload) and negative (non-upload) samples.

**Data volume.** We used 100 consecutive days of online logs to generate over 30 billion training instances. Treatments were treated as short video touchpoints within the recommendation system that could potentially promote creation or increase activity, organized into consumption sequences based on user chronological order. The number of treatment candidates reached the billion-level.

**Labels.** The outcome variable $Y$ is a binary upload signal $(0/1)$. To enrich supervision without leaking $Y$, we introduce a proxy signal $Y'$ in the CDP scenario. Specifically, $Y'$ measures the similarity between the user's new work and the candidate treatment according to pre-defined strategies and rules. By design, $Y'$ acts as an adversarially constrained, observable intermediary proxy that provides the necessary path information for front-door identification.

**Positive and Negative sampling strategy.** Positives are users who uploaded contents. The overall positive-negative sample ratio was controlled at 2:3 to ensure balanced classification and statistical power. We ensure the interaction volume of negatives, quantified by the length of collect and like sequences. The upload activity distribution of negatives is aligned with that of the positives. The specific sampling strategy is shown in Table 1.

**Streaming training.** We construct a Hive table in a list-wise format every hour. Stratified negative samples are used to control computational cost while preserving representativeness, and feature generation is performed via the Kaiworks FG pipeline. During inference process,

Table 1: The specific positive and Negative sampling strategy.

| $L$ | POSITIVE | NEGATIVE | RATIO |
|---|---|---|---|
| $L < 20$ | 5 billion | 4,440 billion | 0.16% |
| $20 < L < 120$ | 3 billion | 840 billion | 0.54% |
| $L > 120$ | 2 billion | 280 billion | 1.07% |

the fine-ranking model queries the MTA model to obtain attribution labels, which are transformed into point-wise samples with full shuffling for training stability.

### 5.4 CAUSALITY EVALUATION SCHEME

Since randomized controlled trials (RCTs) are not available, the true average treatment effect (ATE) cannot be directly observed. Standard predictive metrics such as AUC only assess discrimination accuracy but fail to measure whether the predicted uplifts correspond to genuine causal contributions. To bridge this challenge, we designed a non-personalized AUUC protocol based on Shapley

value sampling, which provides a principled evaluation of attribution quality on observational data. Appendix §A.4 shows the full non-personalized bucketed AUUC protocol.

## 5.5 EFFECTIVENESS VERIFICATION

**Ablation Studies and Mechanism Analysis.** We ablate the causal pipeline to quantify the contributions of mediator observation and contrastive adaptation. First, a DML counterfactual baseline with propensity correction and privileged inputs achieves an AUC of only $0.6498$ and UAUC of $0.50$, indicating that latent system-level confounding remains unresolved by backdoor adjustment alone. Second, while direct mediator observation through $Y'$ inflates discrimination (AUC $0.97$, UAUC $0.90$), it induces severe outcome leakage and training instability, as evidenced by large loss oscillations and failure to converge (Fig. 5(a)). By introducing adversarial learning, ALM-MTA effectively eliminates this leakage and stabilizes convergence, yielding a robust AUC of $0.86$ and UAUC of $0.71$ (Fig. 5(b)). Furthermore, integrating a MoCo-style contrastive learner addresses the sparsity in large-scale treatment spaces by ensuring overlap, which further boosts predictive accuracy to AUC $0.907$ and UAUC $0.825$. Finally, for established attribution categories, the method improves AUC to $0.63$–$0.70$.

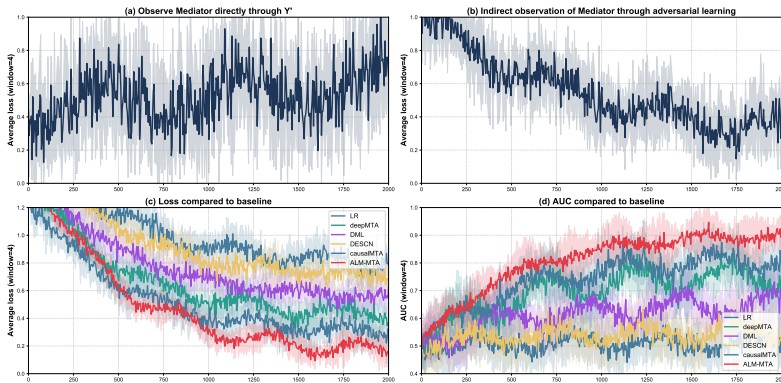

Figure 5: Training dynamics and ablation analysis. (a) Direct proxy observation leads to non-converging oscillation, indicating outcome leakage. (b) Adversarial learning removes leakage, resulting in stable convergence. (c) Loss comparison with baselines, where ALM-MTA shows higher initial loss (due to MTL) but converges effectively. (d) AUC comparison, demonstrating that ALM-MTA achieves superior discrimination against baselines as training progresses.

**Results and discussion.** We evaluated the performance of all methods using AUC, log loss, gAUC and averaged grouped AUUC. Grouped AUUC measures causal validity by ranking predicted uplifts within propensity-stratified treatment-cluster buckets and integrating the uplift curve—larger area indicates better alignment with realized causal gains. Appendix §A.4 shows the full non-personalized bucketed protocol. Logistic regression performs poorly under observational confounding,

Table 2: Methods performance across discrimination, calibration and causal ranking metrics. AUC and gAUC assess discrimination; log-loss reflects calibration; avg AUUC measures grouped causal uplift ranking. ALM-MTA attains the best results on all four metrics.

|          | AUC    | log-loss | gAUC   | avg AUUC |
|----------|--------|----------|--------|----------|
| LR       | 0.5102 | 0.7833   | 0.5011 | 0.4725   |
| deepMTA  | 0.7598 | 0.4108   | 0.6364 | 0.6277   |
| DML      | 0.6498 | 0.5433   | 0.5031 | 0.8421   |
| DESCN    | 0.5492 | 0.6920   | 0.5003 | 0.8429   |
| causalMTA| 0.8167 | 0.2835   | 0.6752 | 0.8493   |
| ALM-MTA  | 0.9070 | 0.1384   | 0.8210 | 0.8686   |

with AUC 0.5102, log loss 0.7833, gAUC 0.5011, and grouped AUUC 0.4725. Deep sequence models such as deepMTA improve discrimination to AUC 0.7598 with lower log loss 0.4108, yet their gAUC 0.6364 and grouped AUUC 0.6277 reveal sensitivity to spurious sequential correlations. DML and DESCN show uneven behavior on ordered, sparse treatments: discrimination remains low (AUC 0.6498 and 0.5492; gAUC 0.5031 and 0.5003) despite higher grouped AUUC around 0.84, indicating unstable calibration and ranking consistency. CausalMTA introduces counterfactual deconfounding and improves to AUC 0.8167, log loss 0.2835, gAUC 0.6752, and grouped AUUC 0.8493, though it still depends on observable confounders. Our ALM MTA attains the best results across all metrics, reaching AUC 0.9070, the lowest log loss 0.1384, gAUC 0.8210, and the highest

grouped AUUC 0.8686, as shown in Table 2 and visualized in Fig. 5(d). The consistent gains across discrimination, calibration, and grouped causal ranking indicate that adversarial mediator observation and contrastive adaptation effectively remove latent confounding and scale to large treatment spaces, yielding reliable and causally faithful attribution.

**Stability Analysis.** Fig. 6 illustrates the stability of attribution estimates across DML, DESCN, CausalMTA and our ALM-MTA. We exclude IR and DeepMTA as they capture only correlational patterns rather than causal effects. We observe that both DML and DESCN exhibit high sensitivity to random initialization, with markedly divergent bimodal pxtr distributions across training runs, leading to inconsistent causal explanations. This instability underscores their inability to provide reproducible causal insights in large-scale recommendation settings. CausalMTA reduces this variance to some extent, yielding more consistent distributions and thus partially improving attribution stability. By contrast, ALM-MTA achieves the most stable behavior: its pxtr distributions remain highly consistent across seeds, indicating that the causal attributions it generates are reproducible and robust. This confirms that the adversarial mediator design and stability-oriented training procedure effectively mitigate randomness in estimation, yielding stable and interpretable causal attributions.

## 5.6 ONLINE EVALUATION AT REAL-WORLD PRODUCTION SCALE

We deployed ALM-MTA in real-world production, where its multi-touch uplift labels are consumed by fine-ranking training and online reweighted serving under fully aligned policies. We evaluate four metrics to ensure causal coherence and reproducibility: *supply-side gains*, *scalability*, *coverage*, and *attribution accuracy*.

**Supply-side gains with steady platform health.** Our first online experiment confirms that the proposed method robustly improves key supply-side metrics. We observe a direct increase in the core active creator population, with Daily Active Users (DAU) rising by $+0.04\%$. The addressable scale of these creators also expands by $+0.119\%$. These gains are mediated by stronger creation-intent signals, as theorized. Engagements with production-related prompts ($+0.292\%$), publication-focused prompts ($+0.789\%$), and social-driven prompts ($+0.465\%$) all increase significantly. This strengthening of the mediator stage ($M$) leads to downstream increases in producer WAU ($+0.566\%$) and trend-following creators ($+0.555\%$), alongside a $+0.065\%$ improvement in content diversity. Importantly, these targeted improvements do not negatively impact overall platform health. Macro-level indicators such as active devices ($-0.010\%$), per-user time spent ($+0.018\%$), and per-device time spent ($+0.010\%$) fluctuate only negligibly. This provides strong evidence that our front-door and adversarial-mediator architecture effectively strengthens the intended $T \rightarrow M \rightarrow Y$ causal chain while preventing reliance on spurious correlations.

**Scalability in larger treatment spaces.** Under heavier load and larger treatment spaces, the method preserves positivity and remains stable at system scale. Producer WAU and author-open scale increase by $+0.145\%$ and $+0.179\%$, respectively. The CDP indicators also rise significantly, with production traffic efficiency at $+0.504\%$ and uploads per user at $+0.569\%$. Stronger consumption-side signals (strict-bandwagon VV $+0.424\%$, broad-bandwagon plays $+0.249\%$) validate the maintenance of overlap by contrastive front-door marginalization. Platform health remains within a narrow fluctuation band (active devices $+0.035\%$, overall time $+0.002\%$), and faster interaction scenarios show enhanced comment dwell ($+0.128\%$), indicating that expanding the treatment space does not come at the expense of platform health.

**Coverage and multi-touch depth.** Under a unified threshold of $0.54$, upload coverage increases from $0.39$ to $0.58$ (up to $1.49\times$), while the average number of attributed touchpoints per covered upload rises from $2.93$ to $7.31$ (up to $2.49\times$). These results indicate that, in high-cardinality treatment spaces, an attribution framework incorporating front-door identification and contrastive learning can substantially broaden the coverage of explainable uploads and increase touchpoint-level explanatory granularity. The resulting higher label density directly improves supervision quality for uplift-oriented re-ranking and incentive allocation.

**Attribution accuracy.** Under the 'matching and activation' criterion, we assess causal links between touchpoints and uploads. Compared with the method currently deployed online (Precision $4.32\%$, Recall $7.95\%$), our method achieves substantial improvements under the same evaluation: Precision increases to $21.88\%$, and Recall increases to $11.67\%$. Moreover, in the high-confidence segment (top-score bucket), Precision reaches $54\%$, indicating strong discriminability and calibration in the head region of the scoring function. Overall, ALM-MTA produces attribution labels with markedly

higher accuracy and practical utility, providing more reliable supervision for downstream uplift re-ranking and incentive allocation.

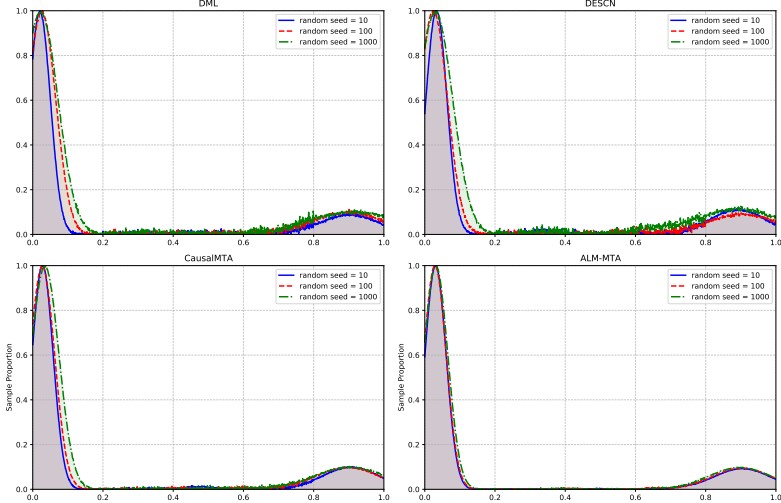

Figure 6: Attribution stability analysis across random seeds. We compare the distribution of attribution scores ($pxtr$) under identical training data but varying random seeds (10, 100, 1000). Baselines (DML, DESCN) exhibit dispersed distributions across seeds, indicating high sensitivity to initialization. In contrast, ALM-MTA demonstrates highly consistent overlap, confirming superior reproducibility and robustness against random factors.

Overall, the online results across these metrics are consistent with offline evaluation and ablation studies. These findings validate the effectiveness of our method's deconfounding strategy and yield attribution labels that are immediately actionable for production ranking and incentive design. The Appendix §A.6 presents diverse and cross-domain attribution case studies to demonstrate the universality of our causal attributions.

## 6 CONCLUSION AND FUTURE WORK

**Conclusion.** We present ALM-MTA, an extensible and practical causal attribution framework for consumption-drives-production platforms. The method (i) casts attribution as a front-door identifiable problem with a latent mediator that transmits the causal pathway from consumption to upload, (ii) renders the mediator observable via an adversarially constrained proxy $Y'$ that is invariant to the outcome $Y$, coupled with contrastive learning, and (iii) ensures scalability and positivity in high-cardinality treatment spaces by restricting front-door marginalization to contrastively identified high-match consumption–upload pairs. Together, these elements remove latent confounding without outcome leakage and yield reproducible, individualized uplift labels with materially higher actionable coverage and label density at production scale. Empirically, ALM-MTA improves predictive accuracy and grouped causal ranking offline and delivers measurable gains to creator-ecosystem metrics online while preserving overall platform health.

**Future work.** (i) Threshold analysis to determine the optimal balance between coverage and precision, thereby maximizing attribution efficiency; (ii) Relaxing the independence assumption via enhanced training architectures (e.g., multi-head attribution) and leave-one/two-out inference to capture higher-order dependencies; and(iii) Improving attribution explainability by elucidating the causal mechanisms linking consumption to production and extending the framework to broader feedback and consumption loops.

## ACKNOWLEDGEMENTS

We thank Huidong Bai for helpful discussions. Authors Yuguang Liu, Luyao Xia and Hu Liu contributed equally to this work.

REPRODUCIBILITY STATEMENT

For community verification and reproducibility, we have released and open-sourced the full codebase in the link [1]. In addition, we will release a de-identified version of the CDP dataset together with the preprocessing scripts in link of github later, so that the complete workflow (from data construction to evaluation) can be fully reproduced. Model design and identification are specified in Sec. 4, covering IPW reweighting, the ITE head, and the adversarial proxy–mediator used for front-door identification (Sec. 4.1). Architectural complexity and compute (FLOPs) are reported in Sec. 5.2 and matched by our configs. Data construction and preprocessing (streaming training, positive/negative sampling, and the proxy label $Y'$) are documented in Sec. 5.3. Offline metrics include AUC, log-loss, gAUC, and propensity-stratified grouped-AUUC; the AUUC definition and the full non-personalized bucketed protocol are given in Sec. 5.4 and App. A.4, and are reproduced in our evaluation toolkit. To demonstrate reproducibility, we report multi-seed stability; our method consistently regenerates the stability plots in Fig. 6.

THE USE OF LARGE LANGUAGE MODEL

We used large language models (LLMs) only to aid or polish the writing; no content generation, ideation, or data analysis was performed by LLMs.

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

## A APPENDIX

### A.1 PROXY-BASED JUSTIFICATION OF THE MEDIATOR $Y'$

In our CDP setting, the latent mediator $M$ represents an "inspiration" or creative-intent signal that connects content consumption to subsequent uploads. Since $M$ is not directly observed, we follow the proxy identification literature Miao et al. (2018b); Tchetgen Tchetgen et al. (2024b) and use a proxy variable $Y'$ as a surrogate for $M$. The proxy $Y'$ is not an arbitrary auxiliary label; it is defined by strong business rules that reflect how consumption can inspire production in our platform. Concretely, $Y'$ indicates whether a consumed video and a subsequently uploaded video (i) use the same music or template, or (ii) are highly similar in an embedding space, capturing similar themes or storylines. Before this work, our production system already used $Y'$ directly as a heuristic link from consumption to production. Empirically, strict element-level matching (same music/template) covers about 7% of daily uploads, and broader semantic matching contributes roughly another 40%, so the proxy-based inspiration paths account for a substantial portion of creator activity.

Following the proxy-identification framework of Miao et al. and Tchetgen Tchetgen et al., an effective proxy for an unobserved mediator should satisfy three key conditions: (i) a *relevance* condition, meaning the proxy is strongly correlated with the latent mediator $M$; (ii) an *exclusion* condition, meaning that conditional on $M$ the proxy does not have a direct path to the outcome $Y$; and (iii) an *identifiability* condition, meaning the proxy exhibits sufficient variation to capture changes in $M$ and allow the causal effect to be identified. In our design, the business rules underlying $Y'$ ensure the relevance and identifiability conditions: $Y'$ is explicitly constructed to track changes in "inspiration" (e.g., reusing music/templates or themes), and the high coverage figures above indicate rich variation across users and content.

To approximate the exclusion restriction and avoid introducing a new shortcut path from $Y'$ to $Y$, we employ *adversarial mediator learning*. The mediator branch is trained to predict $Y'$, while a discriminator simultaneously tries to predict $Y$ from the mediator representation. The mediator network is optimized (via a gradient-reversal layer) to make this prediction impossible, effectively suppressing any information about $Y$ that can be directly recovered from the learned

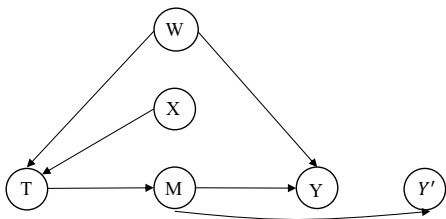

Figure 7: Minimum DAG.

mediator. In this way, the raw proxy $Y'$ is used to construct a "clean" mediator representation that remains strongly related to $M$ but has minimal direct predictive power for $Y$.

Finally, we connect this proxy–mediator construction to the front-door criterion discussed in Section 3. Our causal graph (Figure 2) involves variables $(X, W, T, M, Y)$, where $X$ are observed covariates, $T$ are consumed touchpoints, $M$ is the latent "inspiration" mediator, $Y$ is the upload event, and $W$ denotes unobserved system-level confounders (e.g., latent intent or social influence). The minimum DAG graph is as Fig.7. The standard front-door identification of the effect of $T$ on $Y$ via $M$ relies on three conditions: (i) all directed paths from $T$ to $Y$ go through $M$ (path containment); (ii) there is no uncontrolled confounder affecting both $M$ and $Y$, i.e. $M \perp\!\!\!\perp Y \mid T, X$ (no $M \to Y$ back-door); and (iii) there is no uncontrolled confounder affecting both $T$ and $M$, i.e. $T \perp\!\!\!\perp M \mid X$ (no $T \to M$ back-door).

In large-scale recommendation platforms, Condition (ii) is the most fragile in the raw variables: an unobserved factor $W$ can influence both the mediator $M$ and the outcome $Y$, opening a back-door path $M \leftarrow W \to Y$ and inducing spurious association between $M$ and $Y$ even after conditioning on $T$ and $X$. The combination of the proxy $Y'$ and adversarial mediator learning is designed precisely to approximate this condition at the representation level. The learned mediator representation $\hat{M}$ retains the variation needed to explain $Y'$ and the $T \to M \to Y$ pathway (relevance), while the adversarial objective actively suppresses any shortcut information about $Y$ that leaks through $W$. Thus, when we apply the front-door adjustment in Section 3, $\hat{M}$ should be understood as a "cleaned" surrogate mediator that approximately satisfies the "no $M \to Y$ back-door" requirement in our CDP setting.

The combination of the proxy $Y'$ and adversarial mediator learning is designed precisely to approximate this condition at the representation level. The learned mediator representation $\hat{M}$ retains the variation needed to explain $Y'$ and the $T \to M \to Y$ pathway (relevance), while the adversarial

objective actively suppresses any shortcut information about $Y$ that leaks through $W$. Thus, when we apply the front-door adjustment in Section 3, $\hat{M}$ should be understood as a "cleaned" surrogate mediator that approximately satisfies the "no $M \rightarrow Y$ back-door" requirement in our CDP setting.

**Proxy sensitivity analysis.** We also analyse how the quality of the proxy $Y'$ affects both offline causal metrics and online KPIs. Since $M$ is latent, we cannot directly measure $\mathrm{corr}(Y', M)$. Instead, we rely on an existing "inspiration" module in our CDP system that models $M$ as a function of $(T, Y')$ and is regularly audited by third-party annotators. For each inferred inspiration path, annotators judge whether the consumed content plausibly inspired the upload. From these audits we obtain (i) *path coverage* (the fraction of uploads with at least one inspiration path) and (ii) *precision* (the fraction of paths judged truly inspirational), which together serve as a qualitative proxy for the correlation between $Y'$ and the latent mediator $M$.

To study sensitivity, we start from a very strict definition of $Y'$ and gradually add more proxy signals along business-defined paths (an "add-one" strategy). We consider four nested configurations:

- **Strict element match.** $Y'$ is defined only by exact element matches (same music or template, near-duplicate text). This yields path coverage of about $16.8\%$ with precision close to $100\%$. In this configuration, ALM-MTA achieves grouped gAUUC $\approx 0.8496$ and an uplift in creator DAU of about $+0.201\% + 0.123\% \approx +0.32\%$.

- **Element + semantic match.** We extend $Y'$ to include high semantic similarity in embedding space in addition to exact element overlap. Coverage increases to roughly $28.8\%$ with precision around $82\%$, and grouped gAUUC improves to $\approx 0.8573$, with an additional creator-DAU uplift of about $+0.164\%$.

- **Element + semantic + path.** We further incorporate path-level signals (e.g., collect-and-reuse patterns). Coverage rises to about $39.8\%$ with precision around $88\%$, and grouped gAUUC is $\approx 0.8577$, with an additional DAU uplift of about $+0.105\%$.

- **Full path set.** Finally, we use the full suite of proxy signals. Coverage reaches about $60.4\%$ with precision around $90\%$, and ALM-MTA achieves grouped gAUUC $\approx 0.8686$. In this deployed configuration, the overall uplift in creator DAU is about $+0.514\%$ compared to the rule-based baseline, which can be decomposed into incremental gains of roughly $+0.222\%$ and $+0.160\%$ when the last proxy families are added.

We observe a clear, smooth pattern: when we deliberately weaken the proxy (using only very strict element matches), both gAUUC and creator-DAU gains are reduced but do not collapse. As we enrich $Y'$ with additional semantic and path-level signals, coverage and precision remain high, and both gAUUC and online uplift improve monotonically. This behaviour indicates that ALM-MTA is robust to reasonable variations in proxy quality: as long as $Y'$ remains a meaningful proxy for inspiration, the front-door mechanism continues to deliver consistent causal and business gains.

### A.2 Proof of Unconfoundedness Derivation

We provide the detailed derivation for equation 10. Starting from the interventional definition:

$$\text{upload} = \sum_{t} \text{uplift}_t, \tag{10}$$

$$E[Y \mid do(T = t)] = \sum_{m,x} E[Y \mid M = m, T = t, X = x] \, P(M = m \mid T = t, X = x) \, P(X = x). \tag{11}$$

By the front-door criterion, conditioning on $T$ and $X$ blocks all back-door paths from $M$ to $Y$. Therefore, we can rewrite:

$$E[Y \mid do(T = t)] = \sum_{m,x} f(M = m, T = t, X = x) \, P(M = m \mid T = t, X = x) \, P(X = x), \tag{12}$$

where $f(M, T, X) = E[Y \mid M, T, X]$.

Using observational data, the joint distribution can be reweighted via inverse propensity:

$$P(X, M, T) = \frac{P_{\text{obs}}(X, M, T_{\text{origin}})}{P_{\text{obs}}(T \mid X)}. \tag{13}$$

Substituting, we obtain:

$$E[Y \mid do(T = t)] = \sum_{m,x} f(m, t, x) \frac{P_{\text{obs}}(x, m, T_{\text{origin}})}{P_{\text{obs}}(t \mid x)}. \tag{14}$$

Aggregating across treatments yields the full attribution decomposition:

$$\text{upload} = \sum_{t} E[Y \mid do(T = t)]. \tag{15}$$

This confirms that combining front-door adjustment with IPW produces an "as-if randomized" distribution, enabling unbiased causal attribution directly from observational logs.

### A.3 IMPLEMENTATION DETAILS AND TRAINING RECIPE

**Model architecture.** ALM-MTA is implemented as a multi-task model with a shared backbone encoder $f_\theta(X, T)$ and several task-specific heads: (i) a main head $h^{\text{main}}$ that predicts upload $Y$ and uplift; (ii) a proxy head $h^{\text{proxy}}$ that predicts the proxy $Y'$ and produces a mediator representation $\hat{M}$; (iii) an adversarial head $h^{\text{adv}}$ that predicts $Y$ from $\hat{M}$; (iv) DML-related components that reweight the main loss by the estimated propensity $w(X, T)$; and (v) a contrastive head $h^{\text{ctr}}$ that operates on $(T, Y')$ embeddings for in-batch pairwise contrastive learning. Gradients from explanation-related heads (proxy, adversarial, contrastive) to the backbone are routed using a stop-gradient / gradient-surgery scheme so that uplift prediction remains the primary optimization objective.

**Optimization.** Dense parameters are optimized with Adam (initial learning rate $1 \times 10^{-5}$), and sparse / embedding parameters with Adagrad (initial learning rate $5 \times 10^{-4}$). We apply standard L2 regularization on all trainable parameters. Unless otherwise stated, models are trained with large mini-batches under a data-parallel setup over multiple workers.

**Losses and weighting.** Let $L_{\text{main}}$ denote the main loss on $Y$ (uplift / upload), $L_{\text{DML}}$ the DML-related regularization, $L_{\text{adv}}$ the adversarial loss on $Y$ given $\hat{M}$, $L_{\text{reg}}$ the sum of regularization terms, and $L_{\text{ctr}}$ the in-batch contrastive loss. The total objective is

$$L = L_{\text{main}} + \lambda_{\text{DML}} L_{\text{DML}} + \lambda_{\text{adv}} L_{\text{adv}} + \lambda_{\text{reg}} L_{\text{reg}} + \lambda_{\text{ctr}} L_{\text{ctr}}.$$

In practice, we choose coefficients so that the average loss scales satisfy

$$L_{\text{main}} : L_{\text{DML}} : L_{\text{adv}} : L_{\text{reg}} : L_{\text{ctr}} \approx 1 : 0.6 : 4 : 0.1 : 0.2.$$

Concretely, we use a small base weight (e.g., $10^{-3}$) for the contrastive term and adjust $\lambda_{\text{DML}}$, $\lambda_{\text{adv}}$, and $\lambda_{\text{reg}}$ such that the monitored loss magnitudes fall into the above ratio. This keeps gradient contributions from different objectives within the same order of magnitude, which we find sufficient for stable training at industrial scale.

Moreover, our empirical tuning guideline is that the different loss terms should have similar magnitude on the training curves (within 1 order of magnitude). Under strong regularization and at industrial scale, we find that the method is quite insensitive to small perturbations of these hyperparameters(see Fig.8).

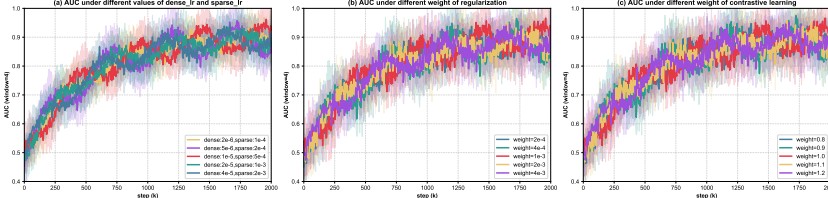

Figure 8: Parameter sensitivity analysis. (a) The changes in the learning rates of dense and sparse parameters have limited impact on AUC level at convergence, verifying the insensitivity to the lr hyperparameter. (b) The changes in weight of regularization have limited impact on AUC. (c) The changes in weight of contrastive learningloss have limited impact on AUC.

**Training schedule.**     We adopt a lightweight staged schedule:

(i) Backbone warm-up. Train the shared backbone and main head with DML (i.e., $L_{\mathrm{main}} + \lambda_{\mathrm{DML}} L_{\mathrm{DML}}$) until the offline AUC curve becomes stable.

(ii) Proxy co-training. Enable the proxy head and jointly optimize $L_{\mathrm{main}}$ and the proxy loss so that the mediator representation $\hat{M}$ captures $Y'$-related signal, while still treating $L_{\mathrm{main}}$ as the primary objective.

(iii) Adversarial and contrastive annealing. Starting from zero weights, gradually increase $\lambda_{\mathrm{adv}}$ and $\lambda_{\mathrm{ctr}}$ to their target values. This avoids an over-strong adversary or contrastive term at early stages and keeps $L_{\mathrm{main}}$ dominant; in practice we monitor that all loss terms remain within roughly one order of magnitude.

## A.4    OFFLINE CAUSALITY EVALUATION: PROPENSITY–STRATIFIED GROUPED AUUC

We first cluster the high-cardinality touchpoints into approximately $K = 1000$ categories using multimodal embeddings, denoted as $T'$. This reduces sparsity and enables reliable estimation of uplift at the cluster level. For each treatment cluster, we estimate its causal contribution using an unbiased Shapley value estimator. Instead of exhaustively computing all marginal contributions, we apply sampling:

$$\hat{\phi}_i(v) = \frac{1}{K} \Sigma_{k=1}^{K} \frac{v(S^{(l)} \cup \{i\}) - v(S^{(k)})}{p(S^{(k)}|i)}, \tag{16}$$

where $S^{(k)}$ is a randomly sampled subset for participant $i$ in the $k$-th sampling, and $v(\cdot)$ is the model's value function (expected outcome). $p(S^{(k)}|i)$ denotes the probability of sampling the subset $S^{(k)}$ conditioned on including participant $i$. The resulting $\hat{\phi}_i$ is The estimated value of the Shapley value for participant $i$.

To further reduce bias, we stratify users by estimated propensity scores $P(T'|X)$. This ensures that comparisons between exposed and unexposed units within each stratum are balanced, mitigating residual confounding. Finally, the Area Under the Uplift Curve (AUUC) is computed by aggregating Shapley-based uplifts across users and treatments:

$$AUUC = \frac{1}{N} \Sigma_{n=1}^{N} \Sigma_{i=1}^{t} uplift_{ni}, \tag{17}$$

where $uplift_{ni}$ is the simplified Shapley value assigned to treatment $i$ for user $n$. AUUC evaluates whether higher predicted uplifts correspond to higher realized causal gains, thus capturing the causal validity of attribution.

**Goal.**     In non–RCT logs, absolute ATE is unobservable and naive correlational metrics (e.g., AUC) do not validate causal usefulness. We therefore design a *non–personalized bucketed protocol* that approximates causal uplift at the treatment–cluster level and evaluates whether a model's predicted uplifts align with these estimates.

**Setup.**     Let $\mathcal{U}$ be users, $\mathcal{T}$ be consumed touchpoints, and $Y$ the upload outcome. We cluster touchpoints by multimodal content embeddings into $K$ treatment clusters $\mathcal{C} = \{1, \ldots, K\}$ (typical $K \approx 10^3$). For each user $i$ and cluster $c$, define an exposure indicator $E_{i,c} \in \{0, 1\}$ (user $i$ consumed any item from cluster $c$ in the window). Let $X_i$ denote user/context covariates. A model under test outputs predicted uplifts $\hat{u}_{i,c}$.

**Propensity estimation & stratification (PSM-style debiasing).**     To mitigate user-preference bias that invalidates purely data-driven ground truth, we estimate exposure propensity for each $(i, c)$ pair,

$$e_{i,c} = P(E_{i,c} = 1 \mid X_i),$$

using a simple, leakage-free classifier (e.g., logistic regression). We stratify $(i, c)$ pairs into $B$ propensity buckets by quantiles of $e_{i,c}$ (e.g., $B = 10$). Within each bucket $b$, units have comparable exposure likelihoods, reducing static/dynamic preference bias and enabling fair comparisons.

**Shapley-based uplift labeling within buckets.**     Within each bucket $b$, we estimate per-pair causal contribution using a simplified Shapley sampler that treats treatment clusters as players:

$$\hat{\phi}_{i,c}^{(b)} = \frac{1}{L} \sum_{\ell=1}^{L} \frac{v_i\big(S^{(\ell)} \cup \{c\}\big) - v_i\big(S^{(\ell)} \setminus \{c\}\big)}{\Pr\big(S^{(\ell)} \mid c\big)}, \tag{18}$$

where $S^{(\ell)}$ is a random subset of clusters preceding $c$ in a random permutation and $v_i(\cdot)$ is the user-level value function (expected upload under the subset). This produces *de-biased, bucket-conditional* uplift labels that are more robust than raw Shapley alone.

**Grouped AUUC.** For each bucket $b$, sort pairs $(i, c)$ by model scores $\hat{u}_{i,c}$ in descending order, obtaining the sequence $\{(i_{b,(1)}, c_{b,(1)}), \ldots, (i_{b,(N_b)}, c_{b,(N_b)})\}$. Define the cumulative causal gain

$$U_b(k) \;=\; \sum_{j=1}^{k} \hat{\phi}^{(b)}_{i_{b,(j)}, c_{b,(j)}}, \quad k = 1, \ldots, N_b,$$

and normalize by the total uplift $Z_b = \sum_{j=1}^{N_b} \left| \hat{\phi}^{(b)}_{i_{b,(j)}, c_{b,(j)}} \right|$. The bucket-level AUUC is the Riemann sum of the normalized curve:

$$\text{AUUC}_b \;=\; \frac{1}{N_b} \sum_{k=1}^{N_b} \frac{U_b(k)}{Z_b}. \tag{19}$$

Finally, the *grouped AUUC* aggregates across buckets with size weights $w_b = N_b / \sum_{b'} N_{b'}$:

$$\text{gAUUC} \;=\; \sum_{b=1}^{B} w_b \, \text{AUUC}_b. \tag{20}$$

Higher gAUUC indicates better alignment of predicted uplifts with bucket-conditional causal gains.

**Why this improves offline evaluation.** (i) **De-biasing:** Shapley alone is data-driven and suffers from user-preference bias; propensity stratification (PSM-style) removes major static/dynamic bias, making the Shapley labels usable as *proxy ground truth*. (ii) **Positivity:** Bucketing enforces overlap by comparing units with similar propensities, stabilizing estimates in high-cardinality treatment spaces. (iii) **Comparability:** Ranking within buckets prevents models from exploiting exposure frequency artifacts and focuses evaluation on causal ordering quality.

### A.5 Supplementary Experiments on Public Criteo Dataset

Criteo (Criteo Attribution Modeling for Bidding Dataset) datasetDiemert Eustache, Meynet Julien et al. (2017) represents a sample of 30 days of Criteo live traffic data. Each line corresponds to one impression (a banner) that was displayed to a user. For each banner we have detailed information about the context, if it was clicked, if it led to a conversion and if it led to a conversion that was attributed to Criteo or not. Data has been sub-sampled and anonymized so as not to disclose proprietary elements. The average touchpoint sequence length is 2.68, the longest touchpoint sequence length is 880.

Using cost (bucketing value, which makes modeling easier to converge and further reduces information leakage) as $Y'$, adversarial learning is used together with impressions to eliminate shortcuts from $T->Y$ and $Y->Y'$. The variables after adversarial learning are used as observations of M; the longest sequence length is set to 16.

**AUC and logloss:** To ensure stability, the model incorporates adversarial learning, resulting in a larger initial loss and thus a higher logloss after convergence compared to causalMTA. ALM-MTA models the transformation better (i.e., has a higher AUC). Furthermore, the confidence intervals of ALM-MTA are more stable, primarily due to further optimization of the stability component. To some extent, ALM-MTA exhibits better interpretability and robustness than other baseline models, yielding more stable causal conclusions (see Table.3).

Table 3: AUC and log-loss between ALM-MTA and causalMTA.

| | AUC | log-loss |
|---|---|---|
| causalMTA | 0.9659±0.01 | 0.0517±0.003 |
| ALM-MTA | 0.9729±0.01 | 0.0634±0.002 |

**Budget and CVR:** Because of the introduction of cpp/attribute (adversarial learning leakage prevention) as a front-door observation signal, ALM-MTA is more sensitive to cost/attribute and can achieve better CPA. For CVR, the improvement is limited at 1/2 budget, but results in 1/4, 1/8 and 1/16 budget illustrate that ALM-MTA is more sensitive to cost, especially when the budget is limited (see Table.4).

Table 4: Performance comparison under different budget constraints.

| Model | CPA | | | | Conversion | | | | CVR | | | |
|---|---|---|---|---|---|---|---|---|---|---|---|---|
| | 1/2 | 1/4 | 1/8 | 1/16 | 1/2 | 1/4 | 1/8 | 1/16 | 1/2 | 1/4 | 1/8 | 1/16 |
| DeepMTA | 36.25 | 30.60 | 26.08 | 25.97 | 1372 | 880 | 549 | 289 | 0.1194 | 0.1202 | 0.1236 | 0.1249 |
| CAMTA | 32.61 | 29.73 | 26.05 | 26.25 | 1270 | 864 | 538 | 211 | 0.1127 | 0.1160 | 0.1191 | 0.1166 |
| CausalMTA | 30.34 | 29.52 | 26.45 | 25.47 | 1441 | 976 | 548 | 255 | 0.1247 | 0.1265 | 0.1305 | 0.1283 |
| ALM-MTA | 27.32 | 24.85 | 22.71 | 22.03 | 1503 | 1070 | 557 | 291 | 0.1248 | 0.1271 | 0.1320 | 0.1298 |

A.6 CASES

In the CDP business, users are influenced by the videos they have viewed in the past, which in turn inspires them to create. From the case study, we can see that ALM-MTA captures the causal relationship from video (treatment) to $Y$ (upload behavior) in multiple scenarios(see Fig.9).

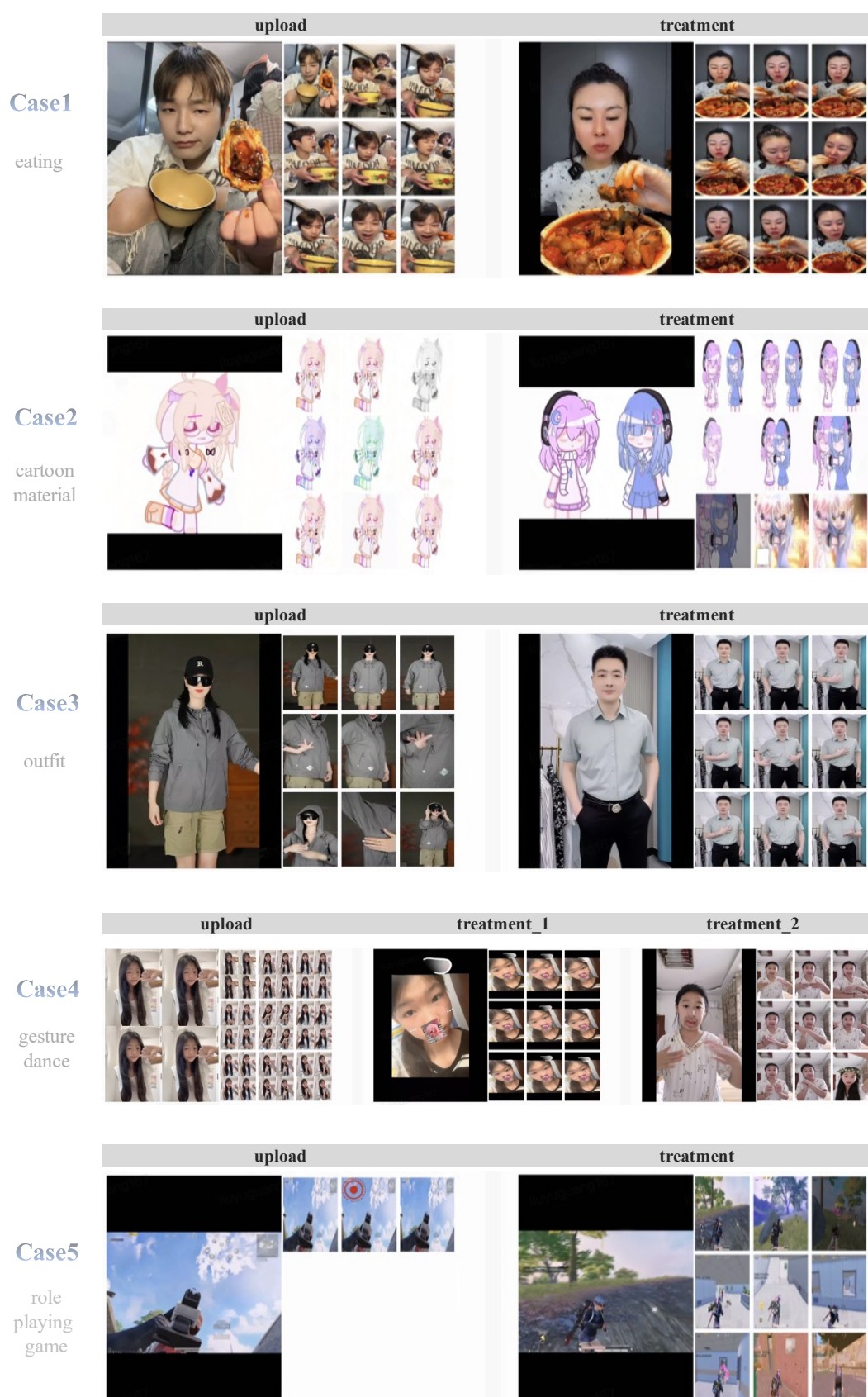

Figure 9: Causal Attribution from Historical Video Views to User Upload via ALM-MTA.

