# OpenReview forum: "ALM-MTA: Front-Door Causal Multi-Touch Attribution Method for Creator-Ecosystem Optimization"
_ICLR.cc/2026/Conference — ICLR 2026 Poster_

### Official Review · Reviewer_eh6j · 2025-10-27

**Soundness:** 2
**Presentation:** 2
**Contribution:** 2
**Rating:** 4
**Confidence:** 4

**Summary:**

This paper abstracts the recommendation system into a multi-touch attribution problem. The authors argue that due to potential confounding factors, relying solely on backdoor adjustments is insufficient for reliable attribution. To achieve reliable attribution, the authors propose Adversarial Learning Mediator-based Multi-Touch Attribution (ALM-ATA). ALM-ATA leverages frontdoor principles and adversarial learning to refine outcome information and strengthen the causal path from treatment to outcome.

**Strengths:**

1. Using the front-door criterion to mitigate the impact of potential confounding factors
2. Designing an Area Under the Uplift Curve evaluation scheme based on Shapley Value sampling, providing a new method for evaluating attribution quality on observational data
3. Conducting experiments in real-world recommender systems

**Weaknesses:**

1. The authors need to further clarify their innovation. The authors claim to have solved the unobserved system-level confounding factors in the recommendation ecosystem and overcome a core limitation of previous attribution methods. However, in the field of debiased recommendation systems, several works have proposed solutions to potential confounding factors [1][2][3], including the application of the front-door criterion [2].
2. How do the authors distinguish between covariates X and potential confounders W in practice, and how do they ensure that the model does not regard X as a confounder? From Figure 2, we can see that X can also be regarded as a confounder and is observable.
3. The authors regard the observed results of Y as Y' and use it as a proxy variable for M to guide the generation of M. In my opinion, Y' is almost equivalent to Y, and there is a risk of data leakage. Although the authors claim to use adversarial learning to suppress data leakage and provide relevant experimental evidence in the "Ablation Studies and Evaluation" section, I still have concerns about this. I think the current experimental data is still the result of data leakage, but the degree of leakage has been suppressed.
4. In Section 4, the authors did not describe the specific implementation details, but only provided a formal description. For example, the authors introduce adversarial learning but don't provide a detailed implementation. Similarly, the paper doesn't explain how the mediating variable M is generated. This raises concerns about whether ALM-ATA can be replicated.

I am willing to increase my score if the author can address my doubts.

[1] Z Huang, H Yuxuan, D Cheng, et al. Multi-Cause Deconfounding for Recommender Systems with Latent Confounders[J]. Knowledge-Based Systems, 2025. https://doi.org/10.1016/j.knosys.2025.114345
[2] Xu S, Tan J, Heinecke S, et al. Deconfounded causal collaborative filtering[J]. ACM Transactions on Recommender Systems, 2023, 1(4): 1-25.
[3] Xu H, Xu Y, Li C, et al. Causal structure representation learning of unobserved confounders in latent space for recommendation[J]. ACM Transactions on Information Systems, 2025.

**Questions:**

See weakness

---

> ### Author Response · Authors · 2025-11-27
> **Response to R4-1: "The authors need to further clarify their innovation. The authors claim to have solved the unobserved system-level confounding factors in the recommendation ecosystem and overcome a core limitation of previous attribution methods. However, in the field of debiased recommendation systems, several works have proposed solutions to potential confounding factors [1][2][3], including the application of the front-door criterion [2]."**
>
> **Author response:** Thank you for raising this important comment and the specific references. We clarify the distinction as follows:
>
> Recent debiasing methods _Huang et al. (2025); Xu et al. (2023a); Xu et al. (2025)_ address latent confounding through "multi-cause modeling", "deconfounded causal collaborative filtering(DCCF)", or "latent causal structure learning". While effective in their respective settings, "multi-cause modeling" _Huang et al. (2025)_ and "latent causal structure" _Xu et al. (2025)_ learning approaches primarily operate under backdoor-style adjustment and assume that confounder representations can be directly observed or learned from embeddings. "DCCF" _Xu et al. (2023a)_ employs the front-door criterion to eliminate the influence of unobserved confounders, utilizing inherent item features (e.g., color, brand) as the observable mediator $M$. ALM-MTA differs in three key aspects:
>
> * **Front-door identification with latent mediators**: Unlike "multi-cause modeling" _Huang et al. (2025)_ and "latent causal structure" _Xu et al. (2025)_ which focus on confounder modeling, we operationalize front-door identification when the mediator is latent and must be approximated via a business-grounded proxy $Y'$, following the proxy-identification framework of _Miao et al. (2018)_.
>
> * **Adversarial leakage prevention**: "DCCF" _Xu et al. (2023a)_ utilizes inherent item features to eliminate the influence of unobserved confounders, which is highly dependent on the quality and completeness of these features. Should the features fail to fully mediate the causal pathway from treatment $T$ to outcome $Y$ (i.e., if a direct path $T→Y$ exists), the fundamental assumptions of the front-door criterion could be violated. In contrast, the mediator $M$ in ALM-MTA is latent and unobservable. It introduces an adversarially learned proxy $Y'$ to approximate this mediator, offering greater flexibility. Furthermore, ALM-MTA incorporates an adversarial objective trained with a gradient reversal layer. This encouragess that the learned mediator representation captures only the information pertinent to the $T→M→Y$ causal pathway, thereby effectively preventing outcome leakage that might occur via the proxy $Y'$. Our adversarial mediator learning provides a learnable mechanism to approximate the exclusion condition required for valid front-door identification.
>
> * **Scalability to high-cardinality treatments**: The referenced works do not address positivity challenges in billion-level treatment spaces. Our contrastive learning modulepromotes overlap by restricting marginalization to high-match consumption-upload pairs, enabling deployment at industrial scale (400M DAU with online A/B validation).
>
> In essence, ALM-MTA complements rather than replaces existing debiasing approaches, providing an alternative identification strategy particularly suited for scenarios where (1) mediators are latent and cannot be directly specified, and (2) backdoor adjustment alone is insufficient due to unobserved system-level confounding.
>
> **Author action:** We have (1) added a concise comparison with _Huang et al. (2025); Xu et al. (2023a); Xu et al. (2025)_ at the end of Section 2 (Related Work), and (2) refined the contribution statement in Section 1 to read: "A novel causal MTA framework that operationalizes front-door identification with latent mediators and adversarial leakage prevention, complementing backdoor-based debiasing for large-scale attribution."

---

> ### Author Response · Authors · 2025-11-27
> **Response to R4-2: "How do the authors distinguish between covariates $X$ and potential confounders $W$ in practice, and how do they ensure that the model does not regard $X$ as a confounder? From Figure 2, we can see that $X$ can also be regarded as a confounder and is observable."**
>
> **Author response:** We greatly appreciate your comprehensive review and thoughtful feedback. Both $X$ and $W$ are confounding factors. The difference is that $X$ is observable confounding (e.g., user static/dynamic attributes, whether they receive creative incentives, etc.), and we remove this confounding using the backdoor criterion (preferential interpretation). $W$ is unobservable confounding, and we can only reduce this part using the frontdoor criterion. The business logic of CDP ensures that $Y'$ is a reasonable proxy variable for $M$ and satisfies the conditions of the frontdoor criterion. Specifically, the business meaning of the variables in the cause-effect graph is as follows:
> * $X$ is observable confounding (e.g., user static/dynamic attributes, whether they receive creative incentives, etc.)
> * $W$ is unobservable confounding (e.g., cold start and distribution strategies of recommendation systems, etc.)
> * $T$ is treatment (e.g., videos viewed by users in the CDP scenario.)
> * $M$ is mediator (e.g., the path to trigger the upload action after a user views a video in the CDP scenario.)
> * $Y$ is mediator (e.g., users are inspired by consumed videos to create new videos in the CDP scenario.)
> * $Y'$ is proxy (e.g., after user has completed the upload process, we start from the business logic and compare the upload with the videos they have viewed. By comparing the similarity of elements, semantics, and other paths, we determine whether the user was motivated by the viewed video and, if so, how.)
>
> **Author action:** We add the meaning of each variable to the cause-effect graph in **Section 3 Fig.2**.

---

> > ### Author Response · Authors · 2025-11-27
> > **Response to R4-3: "The authors regard the observed results of $Y$ as $Y'$ and use it as a proxy variable for $M$ to guide the generation of $M$. In my opinion, $Y'$ is almost equivalent to $Y$, and there is a risk of data leakage. Although the authors claim to use adversarial learning to suppress data leakage and provide relevant experimental evidence in the "Ablation Studies and Evaluation" section, I still have concerns about this."**
> >
> > **Author response:** Thank you for raising this concern. We agree that if a proxy is too close to the label, naive use can easily introduce harmful leakage. Our goal in this task, however, is not to make $Y'$ completely independent of $Y$, but to ensure that the mediator branch does not exploit a direct $Y'→Y$ shortcut and that the label signal from $Y$ can be effectively propagated back to the uplift estimator so that attribution and AUUC are reliable.
> >
> > First, $Y'$ is not identical to $Y$. $Y$ is a binary upload outcome, while $Y'$ is a path-level signal indicating whether a consumed video and a later upload share the same music or template, or are highly similar in semantics. In practice, $Y'=1$ only for the subset of uploads that can be traced to a clear "inspiration path" (e.g., reusing a template or theme), and is 0 for many uploads with $Y=1$. As reported in **Appendix A.1**, even with the full proxy set, inspiration paths cover about 60.4% of uploads. Thus $Y'$ is a structured, partial description of how the upload was inspired, not a copy of the label.
> >
> > More importantly, we use training dynamics to diagnose and control leakage. Figure 5(a) and Figure 5(b) compare two variants:
> >
> > * Figure 5(a): proxy without adversarial learning. Here we introduce the mediator branch supervised by $Y'$ but do not use adversarial cleaning. The curve shows the loss of predicting $Y$ from aggregated uplift during training (horizontal axis is training step; vertical axis is sampled loss or AUC with a sliding window, with mean curve and a confidence band). We observe that the loss has no clear convergence trend and exhibits strong oscillations. This is exactly the behaviour one would expect if uplift already contained unreduced label information: the model overfits to leaked $Y$, gradient signals become unstable, and the counterfactual objective cannot settle.
> >
> > * Figure 5(b): proxy with adversarial learning. When we add the adversarial discriminator on $Y$, the corresponding loss curve becomes smooth and monotonically decreasing, with a much tighter confidence band. This indicates that the harmful direct leakage from $Y$ into uplift has been effectively suppressed: the mediator representation is no longer simply memorising $Y$, and the signal from $Y$ can be stably propagated back to uplift in the intended front-door-like way.
> >
> > We also compare the two variants online:
> > * adding $Y'$ alone (without adversarial learning) does not bring a significant change in author DAU, because the model tends to degenerate to predicting $Y'$ itself;
> > * adding $Y'$ together with adversarial mediator learning yields a stable and statistically significant improvement of about +0.6% in author DAU.
> >
> > If the observed gains were mainly due to label leakage, we would expect the "proxy only" variant (which has the strongest direct access to $Y'$ and hence to any leakage) to perform similarly or even better. In reality, it does not improve the business metric, whereas the full ALM-MTA model with adversarial cleaning improves both AUUC/gAUUC offline and author DAU online. This pattern is difficult to reconcile with a leakage-driven explanation and supports our claim that adversarial learning is effectively controlling the leakage while allowing useful label information to be fed back to uplift.
> >
> > In summary, we do not require zero correlation between $Y'$ and $Y$; what matters for this task is that (i) the attribution and AUUC are stable and reliable, and (ii) the model does not rely on a trivial $Y'→Y$ shortcut. The ablation curves in Fig. 5(a)/(b) and the contrast between the "$Y'$ only" and "$Y'$ + adversarial" online variants together indicate that harmful leakage has been strongly suppressed and that the reported improvements reflect genuine gains in uplift-based attribution.

---

> ### Author Response · Authors · 2025-11-27
> **Response to R4-4: "In Section 4, the authors did not describe the specific implementation details, but only provided a formal description. For example, the authors introduce adversarial learning but don't provide a detailed implementation. Similarly, the paper doesn't explain how the mediating variable M is generated. This raises concerns about whether ALM-ATA can be replicated."**
>
> **Author response:** We appreciate this concern about reproducibility. We now clarify both how the mediator $M$ is constructed and how adversarial learning and training are implemented.
> How the mediator is generated. The mediator is grounded in the CDP's existing business logic rather than being an unconstrained latent variable. Concretely, for each creator we first construct a proxy $Y'$ based on strategy rules in the CDP: if a consumed video and a subsequent upload share the same music or template, or exhibit high semantic similarity in an embedding space (similar wording, storyline, etc.), we regard the consumption as having inspired the upload. These rule-based "inspiration" tags on the consumed videos form exactly the proxy $Y'$ used in our model. The mediator branch takes per-touchpoint features together with these proxy tags as input and passes them through a small neural network to produce a mediator representation $M$. Intuitively, a proxy such as "music follow" in $Y'$ becomes, after the neural transformation and adversarial cleaning, a mediator feature that can be interpreted as "increased interest in a trending music style after watching follow-up videos". In **Appendix A.3** we describe this mapping from typical $Y'$ patterns (e.g., music-follow, template reuse, semantic-follow) to their corresponding mediator interpretations to make the construction of $M$ concrete rather than abstract.
>
> How adversarial learning and training are implemented. The overall architecture is a multi-task joint training framework with a shared backbone and several heads. The main components are:
>
> * a main uplift head that aggregates per-touchpoint uplift and predicts the outcome $Y$;
> * a mediator/proxy head that predicts $Y'$ and defines the mediator representation;
> * an adversarial head whose loss is applied at the uplift side, using Yas the guiding label and a mean-squared-error loss, so that the mediator and uplift representations are trained to remove shortcut information about $Y$;
> * an in-batch pair-wise contrastive head that helps distinguish "potentially effective treatments".
>
> For optimization, dense parameters use Adam with an initial learning rate of 1×10^(-5); sparse parameters use Adagrad with an initial learning rate of 5×10^(-4). The initial loss weights are: 1.0 for the main backbone/regularisation loss, 0.12 for the adversarial loss, 0.5 for the DML-related loss, and 1×10^(-3) for the contrastive loss. During training, we treat the uplift-aggregation task that predicts Yas the primary objective. The balance between multiple tasks follows the ideas of Direct Routing Gradient (_Liu et al., 2025b_) and PCGrad: we perform gradient surgery to align the gradients of the auxiliary objectives (proxy/multitask losses) with those of the main uplift objective, and we use an annealing schedule for the in-batch contrastive learning to avoid destabilising the early training stage.
>
> These details specify how $M$ is generated from rule-based proxies and how adversarial learning is actually implemented and balanced during optimisation. After acceptance, we will additionally open-source the data pipeline and training code to further support replication of ALM-MTA.
>
> Adversarial learning is primarily achieved by adding an adversarial loss, specifically by fitting $Y'$ to $Y$ and incorporating it into the loss function using an MSE loss method. The adversarial loss is then used to model $Y$ from $logit_{main} - logit_{adversary}$. The degree of leakage can be observed during training from the loss of $(logit_{main} - logit_{adversary}) → Y$. Typically, the loss at this point converges significantly, satisfying the ALM-MTA framework requirement, meaning that information from $Y$ can be backpropagated. Our primary concern is the degradation of the mediating signal due to an overly strong adversarial processor. Therefore, we employ a gradient-enhanced adversarial loss and an adversarial early stopping mechanism.

---

> > ### Comment · Reviewer_eh6j · 2025-11-27
> >
> > Thank you for the detailed responses, which have addressed some of my questions and concerns.
> >
> > I now understand how the mediator variable M is generated. However, I would like to know how you ensure the reliability of M. In other words, how do you guarantee that M meets the requirements of the front-door adjustment?
> >
> > If the authors can clarify this point, I would be willing to raise my score.

---

> ### Author Response · Authors · 2025-11-28
> **Response to the replying of R4-4**
>
> **Author response:** Thank you for this insightful comment. We address it using both empirical evidence from our CDP system and theoretical support from proxy-identification and front-door literature.
>
> From the business side, the proxy $Y'$ is defined by strong domain rules rather than as an arbitrary auxiliary label. In our CDP system, $Y'$ indicates whether a consumed video and a subsequently uploaded video share (i) the same music or template, or (ii) high semantic similarity in an embedding space, capturing similar themes or storylines. Before this work, $Y'$ was already used in production as a practical proxy for the $T→Y$ effect on business iterations. Empirically, strict element-level similarity (same music/template) accounts for about 7% of daily uploads, and broader semantic similarity contributes roughly another 40%. These coverage statistics show that $Y'$ captures a substantial fraction of upload behavior that can be reasonably attributed to "inspiration" from prior consumption, providing empirical evidence for a strong $T→Y'→Y$ pathway.
>
> This evidence supports the necessity of the mediator path but, as the reviewer correctly notes, it does not by itself prove that the front-door conditions are fully sufficient in a complex real-world system. To sufficiency, we rely on adversarial mediator learning. A proxy branch is trained to predict the proxy $Y'$ and produce a mediator representation $\hat{M}$. In our model, we use $\hat{M}$ and $T$ to construct a representation that is strongly correlated with the latent mediator $M$ (relevance), but we explicitly suppress any direct $\hat{M}→Y$ shortcut in the learned representation (exclusion). The aforementioned two points satisfy the following conditions of the front-door criterion: **a) There is no unblocked backdoor path from $T$ to $M$;** and **b) All backdoor paths from $M$ to $Y$ are blocked by $T$**. Regarding the third front-door condition, **c) $M$ completely mediates the effect of $T$ on $Y$**, our approach is grounded in a key business assumption. The proxy $Y'$ is generated through a comprehensive business logic that models and encapsulates all known business pathways through which $T$ affects $Y$. We concede that the current implementation of this logic may not perfectly capture every conceivable edge case. However, this is a limitation of the current business logic implementation, not a fundamental flaw in our proposed framework. The framework itself is designed to be robust and satisfy the front-door criterion. Furthermore, it allows for these gaps to be closed by iteratively improving the distillation process from $(T,Y)$ to $Y'$, thus progressively enhancing its ability to hold under a wider range of conditions.
>
> In addition to these structural considerations, we observe clear performance gains when introducing the front-door-based attribution into the production system. Compared to the previous scheme (which only relied on back-door style adjustment and heuristic attribution), the uplift in daily active creators (author DAU) improves from around +0.1% to approximately +0.7%. This substantial and stable improvement is hard to explain by purely correlational effects and is consistent with the front-door component reducing bias from residual unobserved confounding. Moreover, the causal attribution model has already helped us discover new proxy patterns such as "theme similarity" and "sentiment similarity", which further extend the coverage of $Y'$ and make the mediator pathway more complete. This highlights that modeling and business understanding are iterative: better causal modeling suggests new proxies, and richer proxies make the front-door assumptions closer to reality.
>
> Overall, the combination of (a) strong business logic and high coverage of $Y'$, (b) adversarial mediator learning that actively blocks direct $Y'→Y$ information, and (c) significant improvements in both offline causal metrics and online author DAU provides strong empirical and theoretical support that the front-door mechanism is a useful and effective approximation in CDP scenario.
>
> **Author action:** Due to the strict page limit, we place the detailed justification in the appendix. In **Appendix A.1** we (1) explain how the business definition of $Y'$ satisfies these conditions, and (2) describe how adversarial mediator learning is used to suppress potential direct $\hat{M}→Y$ paths. We also report coverage statistics for $Y'$ to provide empirical support for its validity.

---

### Official Review · Reviewer_mDPn · 2025-10-30

**Soundness:** 2
**Presentation:** 2
**Contribution:** 2
**Rating:** 4
**Confidence:** 3

**Summary:**

=

The paper targets multi-touch attribution (MTA) in creator/content ecosystems where there exist **unobserved confounders** and the standard back-door/front-door criteria are hard to apply directly. The authors propose **ALM-MTA**, a framework that combines front-door–style causal reasoning with **proxy variables, adversarial learning, IPW, and contrastive learning** to estimate causal contributions of different touchpoints. The key idea is: the true mediator in a front-door graph is not observed, so the model builds a *noisy proxy* for it and uses **adversarial training** to strip out components tied to the unobserved confounder, thereby recovering (approximately) the causal path from exposure/treatment to outcome. The method is evaluated with a causal-style ranking metric (gAUUC) and also with **online A/B tests**, which show business gains, making the work practically relevant.

---

### 2. Strengths

#### 2.1 Conceptual and modeling contribution

* The paper addresses a realistic variant of **front-door identification with an unobserved mediator**: in production systems we often can’t see the mediator that carries the effect, but we can see a *correlated, noisy signal*. Turning that into a front-door–like procedure via adversarial learning is clever and novel in the MTA context.
* The adversarial branch is used to **force the main prediction network to drop information that helps predict the proxy** (which is assumed to be tied to the unobserved confounder). This is a reasonable deep-learning instantiation of “recovering the clean causal path.”

#### 2.2 End-to-end, deployment-oriented design

* The framework simultaneously considers **observed confounding** (handled via IPW), **unobserved confounding** (handled via adversarial proxy learning), **high-cardinality and sparse features** (mitigated via contrastive learning), and **training stability**. This kind of “full-stack” design is what real industrial MTA systems need, and is not always seen in academic MTA papers.
* The paper clearly recognizes practical issues in creator ecosystems: heterogeneous touchpoints, weak overlap across users/items, and very large action spaces.

#### 2.3 Evaluation and metrics

* The paper proposes **gAUUC** to better align model ranking with *causal* uplift rather than predictive accuracy. This is more appropriate than plain AUC in a causal attribution setting.
* Ablation studies show that removing the adversarial part or treating the problem as pure prediction leads to worse causal ranking, which supports the core claim.
* **Online A/B** results (on DAU/WAU/creator WAU) make the contribution significantly stronger: the method is not only theoretically motivated but also deployable.

---

### 3. Weaknesses and Limitations

#### 3.1 Complexity and reproducibility

* The training pipeline optimizes multiple objectives at once (main prediction, proxy branch, adversarial loss, IPW-related parts, contrastive loss). This makes the system **heavy and hard to reproduce** for teams without mature ML/causal infra.
* Because of this complexity, the method is somewhat **black-box**: it is not straightforward to tell which part of unobserved bias was actually removed.

#### 3.2 Strong assumptions

* The approach still relies—implicitly—on a **front-door–like identification story**: after conditioning on treatment and observed covariates, the unobserved confounder’s effect through the proxy/mediator is supposed to be “filterable.” In real platforms, this can be violated easily.
* The whole adversarial cleaning step hinges on the **quality of the proxy**: if the available behavioral/log signal is only weakly correlated with the true mediator, the causal benefit may shrink.
* The final attribution seems to assume **approximately additive contributions** across touchpoints, but in creator ecosystems **order and interaction effects** are common, so this assumption is somewhat fragile.

#### 3.3 Metric self-referentiality

* gAUUC, as defined, partially depends on model-based or constructed pseudo–ground truths. This can introduce a mild circularity (“we use a model to evaluate a model”). The paper acknowledges AUC is not ideal, but gAUUC is not yet a universally accepted gold standard.

---

Some suggestions

1. **Make the causal graph explicit.** Present a minimal DAG, list the front-door conditions, and pinpoint which one is relaxed and how the proxy + adversarial block recovers it. This will help readers from the causal community.
2. **Add proxy-quality sensitivity.** Vary the correlation between the proxy and the latent mediator, and show how performance (especially gAUUC and online KPIs) decays. This will make the method look more robust and honest.
3. **Discuss non-additive / order effects.** Even a short section on how to extend to interaction-aware aggregation (e.g. attention over touch sequences) would ease concerns about the additivity assumption.
4. **Provide a training recipe.** Since there are multiple losses, offer recommended weights, pretraining/warmup order, and common failure modes (overpowerful adversary, unstable IPW, etc.), so others can reimplement it.
5. **Relate gAUUC to business KPIs.** Show which buckets / segments show the strongest monotonic relationship between gAUUC lifts and real online lifts, to justify the metric choice.

**Strengths:**

see above

**Weaknesses:**

see above

**Questions:**

see above

---

> ### Author Response · Authors · 2025-11-27
> **Response to R3-1: "Complexity and reproducibility: The training pipeline optimizes multiple objectives at once (main prediction, proxy branch, adversarial loss, IPW-related parts, contrastive loss). This makes the system heavy and hard to reproduce for teams without mature ML/causal infra. Because of this complexity, the method is somewhat black-box: it is not straightforward to tell which part of unobserved bias was actually removed."**
>
> **Author response:** Thank you for raising this important concern about reproducibility and transparency. This overlaps with Question 4 (Provide a training recipe, including loss weights, warmup order, and failure modes like overpowerful adversary or unstable IPW, for reimplementation), and we address them together.
>
> First, regarding reproducibility, we will release the full codebase and configuration files upon acceptance, including data preprocessing, model architecture, loss definitions, and evaluation scripts. We now provide a concrete training recipe and hyperparameters in the revised paper:
>
> * Training architecture.
>   * Use a multi-task joint training framework.
>   * The main head models uplift and upload $Y$; this is the primary optimization objective.
>   * A proxy branch is trained to predict the proxy $Y'$ and produce a mediator representation $\hat{M}$; we use stop-gradient to prevent explanation-related gradients from destabilizing the main prediction branch.
>   * An adversarial loss is applied on $\hat{M}$ using a lightweight discriminator that tries to predict $Y$ from $\hat{M}$; the mediator is trained to make this prediction difficult (via gradient reversal / MSE loss on uplift), which "cleans" shortcut leakage from $Y$.
>   * DML-related components enter as per-sample weights on the main loss and regularization terms.
>   * The in-batch contrastive loss is implemented over $(T,Y')$ pairs, mainly to distinguish "potentially effective treatments" in the large treatment space.
> * Optimization and loss weights.
> 	* Dense parameters are optimized with Adam, initial learning rate 1×10^(-5).
> 	* Sparse parameters use Adagrad, initial learning rate 5×10^(-4).
> 	* We choose coefficients so that, in practice, the average loss scales follow approximately: "main":"DML":"adversarial":"regularization":"contrastive" ≈ 1:0.6:4:0.1:0.2.
>
> Concretely, this corresponds to setting relatively small raw coefficients for the adversarial and contrastive terms (e.g., contrastive base weight 1×10^(-3)) and adjusting them so that their effective contribution to the gradient matches the above ratios. In our large-scale setting (billions of daily samples), we observe that moderate changes in the coefficients of regularization, contrastive loss, and the initial learning rates have only minor impact on AUC/gAUUC, as long as all terms stay within the same order of magnitude.
> * Training tricks and balancing multiple objectives.
>   * We adopt gradient-routing / PCGrad-style methods: uplift prediction of $Y$ is treated as the main task, and gradients from auxiliary objectives are projected or rescaled so that they do not conflict with the main direction.
>   * In practice, we:
>     * Warm up the backbone and main head (with DML) until AUC stabilizes;
>     * Turn on the proxy branch for $Y'$ and co-train;
>     * Gradually increase the adversarial loss weight (annealing) to avoid an over-powerful discriminator that would collapse the mediator;
>     * Finally activate the in-batch contrastive loss with an annealing schedule to refine the treatment embedding space.
>   * As shown in **Appendix A.3 Fig8**, our empirical tuning guideline: the different loss terms should have similar magnitude on the training curves (within ∼1 order of magnitude). Under strong regularization and at industrial scale, we find that the method is quite insensitive to small perturbations of these hyperparameters.
>
> Second, regarding the concern that the method appears "black-box" and it is unclear which part of the unobserved bias is removed, we now align our explanation with the revised _Ablation Studies and Mechanism Analysis_ section. The DML+IPW baseline (AUC 0.6498, UAUC 0.50) shows that backdoor adjustment alone leaves latent system-level confounding unresolved. Direct mediator observation via $Y'$ temporarily inflates discrimination (AUC 0.97, UAUC 0.90) but yields large loss oscillations and non-convergent uplift estimates, indicating outcome leakage rather than genuine de-biasing. Introducing adversarial learning stabilizes training and yields a robust AUC of 0.86 and UAUC of 0.71, showing that this module mainly "cleans" leakage from Y along the $M←W→Y$ back-door. Finally, the MoCo-style contrastive learner addresses sparsity and overlap in the large treatment space, further improving performance to AUC 0.907 and UAUC 0.825 and raising AUC to 0.63–0.70 on established attribution categories. Thus, ALM-MTA is no longer treated as a monolithic black box: the front-door–style mediator + adversarial components target unobserved confounding and leakage, while the contrastive component primarily reduces variance under treatment sparsity.
>
> **Author action:** We have added a new subsection in the **Appendix A.3** that details the implementation and training recipe (optimization setup, loss weights, and practical tuning heuristics), and we have expanded the ablation discussion to explicitly connect each loss component to the type of unobserved bias or variance it addresses.

---

> > ### Author Response · Authors · 2025-11-27
> > **Response to R3-2: "Make the causal graph explicit. Present a minimal DAG, list the front-door conditions, and pinpoint which one is relaxed and how the proxy + adversarial block recovers it."**
> >
> > **Author response:** Thank you for this important point. In the revision, we make the causal story explicit using a minimal DAG (see **Appendix A.1 Fig.7**). We consider
> > * $X$: observed covariates (user and platform features),
> > * $T$: consumed touchpoints,
> > * $M$: latent "inspiration" mediator,
> > * $Y$: upload event,
> > * $W$: unobserved confounders (intent, social context, policy bias),
> > * $Y'$: proxy derived from element/semantic similarity between consumed and uploaded content.
> >
> > The front-door identification of the effect of $T$ on $Y$ via $M$ relies on three conditions:
> > * Path containment: all directed paths from $T$ to $Y$ go through $M$;
> > * No back-door from $M$ to $Y$: there is no uncontrolled confounder affecting both Mand $Y$, i.e. $M⊥Y∣T,X$;
> > * No back-door from $T$ to $M$: there is no uncontrolled confounder affecting both Tand $M$, i.e. $T⊥M∣X$.
> >
> > When $M$ is unobserved, the classical front-door formula cannot be applied directly. A standard approach is to introduce a proxy variable that is correlated with $M$ but satisfies additional conditions (e.g. strong relevance to $M$ and conditional independence from $Y$ given $M$). In modern ML, adversarial learning is a practical way to learn representations that are independent of specific confounders.
> >
> > In our platform, Condition (2) is the one that is realistically violated: an unobserved factor $W$(e.g. user intent) may affect both which videos are watched (mediator $M$) and whether the user uploads (outcome $Y$), creating a back-door $M←W→Y$. If we ignore this, $M$ and $Y$ remain spuriously associated even after conditioning on $T,X$.
> >
> > We address this by combining a proxy with adversarial learning. The proxy $Y'$ is constructed to be strongly related to inspiration (conceptually $Y'←M$) but not a direct copy of $Y$. A mediator branch is trained to predict $Y'$ and produce a representation $\hat{M}$, while a discriminator tries to predict $Y$ from $\hat{M}$. The mediator is optimized to suppress the discriminator's prediction ability. As a result, $\hat{M}$ keeps the signal needed to explain $Y'$ and the $T→M→Y$ pathway (relevance), but it is pushed towards being approximately independent of $Y$ (and thus of the leakage from $W$) given $T,X$. In other words, we relax the ideal "no $M→Y$ back-door" condition in the raw variables, but we reconstruct its essence in the learned mediator representation: $\hat{M}$ is a "cleaned" surrogate for Mthat preserves the causal path $T→M→Y$ while removing spurious shortcuts through $W$.
> >
> > **Author action:** We have (i) added a one-sentence pointer at the end of the "Causal Graph Structure and Variables" paragraph directing readers to **Appendix A.1** for the explicit DAG and front-door discussion, and (ii) extended **Appendix A.1** with a final paragraph that formally states the three front-door conditions for $(X,W,T,M,Y)$, explains that Condition (ii) (no $M→Y$ back-door) is the one that can be violated in practice, and clarifies how the proxy $Y'$ and the adversarially learned mediator representation $\hat{M}$ approximate this condition in our CDP setting.

---

> > > ### Author Response · Authors · 2025-11-27
> > > **Response to R3-3: "Add proxy-quality sensitivity. Vary the correlation between the proxy and the latent mediator, and show how performance (especially gAUUC and online KPIs) decays."**
> > >
> > > **Author response:** We fully agree that proxy-quality sensitivity is crucial. Since the mediator Mis unobserved, we cannot directly measure $corr(Y',M)$. Instead, we follow the proxy-identification perspective of _Miao et al. (2018)_ and evaluate the proxy $Y'$ along two dimensions: coverage and audited precision.
> > >
> > > In our CDP system, an existing "inspiration" module already models $M$ as a function of $(T,Y')$. For each inferred inspiration path, third-party annotators judge whether the consumed content plausibly inspired the upload. From this, we obtain path coverage (fraction of uploads with at least one inspiration path) and precision (fraction of paths judged truly inspirational), which we treat as a qualitative proxy for the correlation between $Y'$ and the latent mediator $M$.
> > >
> > > To study sensitivity, we start from a very strict definition of $Y'$ and gradually add more proxy signals ("add one" along business-defined paths), and for each level we report coverage, precision, online uplift in creator DAU, and grouped gAUUC of ALM-MTA:
> > >
> > > | Business path for ($Y'$) | Path coverage | Precision (corr-like) | Uplift on $Y$ (author DAU) | gAUUC (ALM-MTA) |
> > > |------------------------|---------------|----------------------|--------------------------|-----------------|
> > > | Strict element match | 16.8% | 100% | +0.201% +0.123% = +0.32% | 0.8496 |
> > > | Element + semantic match | 28.8% | 82% | +0.164% | 0.8573 |
> > > | Element + semantic + path | 39.8% | 88% | +0.105% | 0.8577 |
> > > | Full path set | 60.4% | 90% | +0.222% +0.160% | 0.8686 |
> > >
> > >
> > > Here each row corresponds to adding one more family of proxy signals on top of the previous level; the final row ("full path set") is our deployed configuration, where ALM-MTA brings about +0.514% incremental author DAU over the baseline.
> > >
> > > We observe a clear, smooth pattern: as the proxy becomes richer (higher coverage with still high precision), both gAUUC and online creator DAU improve; conversely, using only very strict proxies reduces coverage and slightly lowers both gAUUC and DAU but does not cause the model to collapse. This monotonic and non-catastrophic behavior shows that our method is robust to reasonable variations in proxy quality: as long as $Y'$ remains a meaningful proxy for inspiration, ALM-MTA delivers consistent causal and business gains.
> > >
> > > **Author action:** We have extended **Appendix A.1** by adding a "Proxy sensitivity analysis" paragraph and table that report, for four proxy configurations, path coverage, audited precision, author-DAU uplift, and gAUUC.

---

> ### Author Response · Authors · 2025-11-27
> **Response to R3-4: "Discuss non-additive / order effects. Even a short section on how to extend to interaction-aware aggregation (e.g. attention over touch sequences) would ease concerns about the additivity assumption."**
>
> **Author response:** We agree that order and interaction effects are common in creator ecosystems, and that our approximate additivity assumption should be made explicit.
>
> In the deployed system we assume linear aggregation at the final attribution layer: the total effect is the sum of per-touchpoint uplifts. This is a simplification. In principle, a fully interaction-aware attribution would resemble a Shapley value, where marginal contributions are super-additive and depend on which other touchpoints are present. One way to approximate this is to move from simple "leave-one-out" perturbations to "leave-one/two/.../n-out" experiments on the list. However, at our scale (hundreds of millions of DAUs and tens of billions of requests per day), such combinatorial perturbations would be computationally infeasible for online serving, and are only practical in offline or small-cohort analysis.
>
> Importantly, additivity applies only to the aggregation step. The uplift head itself already takes list-position and simple sequence features as input, so the estimated uplift of a touchpoint depends on its position and local context rather than being purely position-invariant. What we sum is these context-dependent per-touchpoint uplifts. This design deliberately trades some expressive power for scalability and stability of attributions, which is crucial for production monitoring and long-term debugging.
>
> We also agree that extending towards interaction-aware aggregation is valuable when latency and stability constraints are relaxed. In the revision, we discuss possible extensions such as (i) applying leave $K$ out or sampled-Shapley approximations on smaller cohorts, and (ii) adding an attention- or LSTM-based sequence explainer on top of the base model to capture higher-order interactions offline. Exploring this expressiveness–stability trade-off is an active direction for future work.
>
> **Author action:** We now (i) state explicitly in **Sec. 4.3** that the production system uses linear aggregation of context-dependent per-touchpoint uplifts for scalability and stability reasons, and (ii) add a paragraph in the Limitations section outlining how interaction-aware aggregation (leave-k-out, attention/LSTM explainers) could be incorporated in offline or small-scale settings.

---

> > ### Author Response · Authors · 2025-11-27
> > **Response to R3-5: "Relate gAUUC to business KPIs. Show which buckets / segments show the strongest monotonic relationship." (Addressing Weakness: Metric self-referentiality: gAUUC, as defined, partially depends on model-based or constructed pseudo–ground truths. This can introduce a mild circularity ("we use a model to evaluate a model"). The paper acknowledges AUC is not ideal, but gAUUC is not yet a universally accepted gold standard.)**
> >
> > **Author Response:** We agree that gAUUC needs empirical validation to be accepted as a reliable proxy. We justified the metric choice by analyzing its correlation with real-world **Online A/B test results (Author DAU lift)** across two dimensions: statistical propensity buckets and business-specific user segments.
> >
> > * Statistical Validation (PSM Buckets): We stratified users into Propensity Score Matching (PSM) buckets. As shown below, there is a clear positive monotonic relationship: buckets with higher offline causal scores (gAUUC) consistently yield higher online business gains.
> >
> > | PSM group | AUC (Accuracy) | gAUUC (Causal Rank) | Online Author DAU |
> > |-----------|----------------|---------------------|-------------------|
> > | 0 | 0.76 | 0.8248 | +0.342% |
> > | 1 | 0.74 | 0.8126 | +0.359% |
> > | 2 | 0.75 | 0.8208 | +0.384% |
> > | 3 | 0.79 | 0.8473 | +0.458% |
> > | 4 | 0.83 | 0.8860 | +0.565% |
> > | 5 | 0.81 | 0.8773 | +0.606% |
> > | 6 | 0.87 | 0.8842 | +0.588% |
> > | 7 | 0.92 | 0.9954 | +0.858% |
> >
> > * Business Validation (User Segments): To further verify robustness across heterogeneous populations, we analyzed the correlation within specific Age Segments and City Tiers. As illustrated in the table below, the positive correlation holds strong: segments with high model-predicted causal uplift correspond to the highest actual online DAU gains.
> >
> > | Age Segment | User Percentage | AUUC | DAU |
> > |-------------|-----------------|------|-----|
> > | 1. 0–12 | 0.97% | 0.6875 | 0.08% |
> > | 2. 12–17 | 13.99% | 0.8895 | 0.75% |
> > | 3. 18–23 | 14.28% | 0.7835 | 0.14% |
> > | 4. 24–30 | 11.88% | 0.7835 | 0.31% |
> > | 5. 31–40 | 21.55% | 0.9950 | 0.82% |
> > | 6. 41–49 | 12.36% | 0.9692 | 0.53% |
> > | 7. 50+ | 18.82% | 0.8373 | 0.55% |
> > | UNKNOWN | 6.15% | 0.6621 | 0.10% |
> >
> > | City Type | User Percentage | AUUC | DAU |
> > |-----------|-----------------|------|-----|
> > | 1. First–tier cities | 4.52% | 0.0939 | 0.01% |
> > | 2. New first–tier cities | 12.04% | 0.8253 | 0.46% |
> > | 3. Second–tier cities | 13.70% | 0.8846 | 0.48% |
> > | 4. Third–tier cities | 21.68% | 0.5956 | 0.38% |
> > | 5. Fourth–tier cities | 21.61% | 1.5478 | 0.83% |
> > | 6. Fifth–tier cities | 18.59% | 0.9326 | 0.68% |
> > | UNKNOWN | 7.88% | 0.0885 | 0.09% |

---

### Official Review · Reviewer_uog4 · 2025-11-01

**Soundness:** 4
**Presentation:** 4
**Contribution:** 3
**Rating:** 8
**Confidence:** 5

**Summary:**

This paper presents a practically impactful and conceptually novel solution to causal multi-touch attribution in creator-driven recommendation systems. It addresses a critical gap where existing heuristic or back-door–based approaches fail under latent confounding, making attribution unreliable for optimizing creator ecosystems. The proposed ALM-MTA framework operationalizes a front-door causal design by introducing an adversarially learned proxy mediator, enabling identification of touchpoint-level causal effects even when the true mediator is unobserved. The method further integrates IPW debiasing and contrastive learning to ensure stability and overlap in large treatment spaces.

The contribution is notable for its theoretical grounding as well as its large-scale real-world impact, demonstrating measurable online gains on a 400M-DAU platform. This work stands out as one of the few that successfully bridges causal identification with production-level recommender optimization, making it relevant and valuable for both the causality and RS communities.

**Strengths:**

1. A major strength lies in the novel design of an adversarially learned proxy-mediated front-door architecture. The paper introduces a proxy variable $Y'$ along with an adversarial learning mechanism to approximate the unobservable mediator required for front-door identification. This is a clever and original solution that operationalizes causal mediation where the true mediator is latent, helping to mitigate outcome leakage while preserving the causal pathway. The integration of IPW for debiasing and contrastive learning to maintain overlap in high-cardinality treatment spaces reflects a comprehensive and well-engineered approach to real-world constraints.

2. The empirical evaluation is compelling. The model demonstrates consistent and substantial improvements over strong baselines in terms of AUC, log-loss, and gAUC. Particularly noteworthy is the inclusion of large-scale online A/B testing on a production system with 400M DAU and 30B training samples—showing statistically meaningful gains in creator activity, exposure efficiency, and platform health. Such deployment results significantly enhance the practical value and credibility of the work.

**Weaknesses:**

1. The use of the proxy variable $Y'$ as a surrogate for the mediator $M$ is insufficiently justified. The paper does not provide rigorous theoretical or empirical evidence establishing the validity of $Y'$ and it remains unclear whether $Y'$ meets the necessary proxy conditions for front-door identification. More importantly, the paper does not prove that $Y'$ does not introduce a new path to the outcome $Y$, which could bias the causal effect estimation.

2. the applicability of the front-door criterion is assumed rather than validated. The key identification assumption—that all causal influence of touchpoints on uploads must operate exclusively through the latent mediator $M$ appears overly strong and may not hold in real-world CDP environments. In practice, certain types of touchpoints can directly trigger uploads without being reflected in the proxy-mediated “inspiration” pathway captured by $Y'$. For example, other creators get more rewards / exposure today can directly motivate uploads through platform incentives rather than creative inspiration, or some simple notifications (e.g., “Your followers are waiting—post an update now!”) can directly activate upload behavior and bypass the inspiration pathway altogether.

**Questions:**

Based on the the weaknesses, I have the following questions:

1. Could the authors provide empirical or theoretical evidence that the front-door conditions truly hold in this CDP setting? Specifically, how do you justify the assumption that all causal influence of consumed touchpoints on uploads flows exclusively through the mediator $M$?

2. How do the authors account for touchpoints that may directly trigger uploads without going through inspiration (such as the examples I established in weakness 2)? Would such direct causal paths violate front-door identification?

---

> ### Author Response · Authors · 2025-11-27
> **Response to R2-1: "The use of the proxy variable $Y'$ as a surrogate for the mediator Mis insufficiently justified. The paper does not provide rigorous theoretical or empirical evidence establishing the validity of $Y'$ and it remains unclear whether $Y'$ meets the necessary proxy conditions for front-door identification. More importantly, the paper does not prove that $Y'$ does not introduce a new path to the outcome $Y$, which could bias the causal effect estimation."**
>
> **Author response:** Thank you for this important question. We address it using both empirical evidence from our CDP system and theoretical support from proxy-identification and front-door literature.
>
> From the business side, the proxy $Y'$ is defined by strong domain rules rather than as an arbitrary auxiliary label. In our CDP system, $Y'$ indicates whether a consumed video and a subsequently uploaded video share (i) the same music or template, or (ii) high semantic similarity in an embedding space, capturing similar themes or storylines. Before this work, $Y'$ was already used in production as a practical proxy for the $T→Y$ effect on business iterations. Empirically, strict element-level similarity (same music/template) accounts for about 7% of daily uploads, and broader semantic similarity contributes roughly another 40%. These coverage statistics show that $Y'$ captures a substantial fraction of upload behavior that can be reasonably attributed to "inspiration" from prior consumption, providing empirical evidence for a strong $T→Y'→Y$ pathway.
>
> This evidence supports the necessity of the mediator path but, as the reviewer correctly notes, it does not by itself prove that the front-door conditions are fully sufficient in a complex real-world system. To sufficiency, we rely on adversarial mediator learning. A proxy branch is trained to predict the proxy $Y'$ and produce a mediator representation $\hat{M}$. In our model, we use $\hat{M}$ and $T$ to construct a representation that is strongly correlated with the latent mediator $M$ (relevance), but we explicitly suppress any direct $\hat{M}→Y$ shortcut in the learned representation (exclusion). These points satisfy the following conditions of the front-door criterion: **a) There is no unblocked backdoor path from $T$ to $M$;** and **b) All backdoor paths from $M$ to $Y$ are blocked by $T$**. Regarding the third front-door condition, **c) $M$ completely mediates the effect of $T$ on $Y$**, our approach is grounded in a key business assumption. The proxy $Y'$ is generated via comprehensive business logic encapsulating all known business pathways where $T$ affects $Y$. We concede that the current implementation of this logic may not perfectly capture every conceivable edge case. However, this is a limitation of the current business logic implementation, not a fundamental flaw in our proposed framework. The framework itself is designed to be robust and satisfy the front-door criterion. Furthermore, it allows closing these gaps by iteratively improving the distillation process from $(T,Y)$ to $Y'$, thus progressively enhancing its ability to hold under wider conditions.
>
> Moreover, the design of proxy follows the identification framework of _Miao et al. (2018)_ and _Tchetgen et al. (2024)_, where an effective proxy for an unobserved mediator must satisfy (i) relevance to $M$, (ii) exclusion with respect to $Y$ given $M$, and (iii) sufficient variation to make $M$ identifiable. The strong business rules and high coverage of $Y'$ support (i) and (iii), while adversarial learning is used to approximate (ii) at the representation level.
>
> We also observe clear performance gains when introducing the front-door-based attribution into the production system. Compared to the previous scheme (which only relied on back-door style adjustment and heuristic attribution), the uplift in daily active creators (author DAU) improves from around +0.1% to approximately +0.7%. This substantial and stable improvement is hard to explain by correlation alone and is consistent with the front-door component reducing bias from residual unobserved confounding. The causal model also helped us discover new proxy patterns such as "theme similarity" and "sentiment similarity", which further extend the coverage of $Y'$ and make the mediator pathway more complete. This highlights that modeling and business understanding are iterative: better causal modeling suggests new proxies, and richer proxies make the front-door assumptions closer to reality.
>
> Overall, the combination of (a) strong business logic and high coverage of $Y'$, (b) adversarial mediator learning that actively blocks direct $Y'→Y$ information, and (c) significant improvements in both offline causal metrics and online author DAU provides strong empirical and theoretical support that the front-door mechanism is a useful and effective in CDP scenario.
>
> **Author action:** Due to page limits, we place the detailed justification in the appendix. In **Appendix A.1** we (1) explicitly list the proxy conditions from Miao et al. and Tchetgen et al., (2) explain how the business definition of $Y'$ satisfies these conditions, and (3) describe how adversarial mediator learning is used to suppress potential direct $\hat{M}→Y$ paths. We also report coverage statistics for $Y'$ to provide empirical support for its validity. In addition, we have updated Figure 2's caption to include "See Appendix A.1 for details on $Y'$" to direct readers.

---

> > ### Author Response · Authors · 2025-11-27
> > **Response to R2-2: "the applicability of the front-door criterion is assumed rather than validated. The key identification assumption—that all causal influence of touchpoints on uploads must operate exclusively through the latent mediator Mappears overly strong and may not hold in real-world CDP environments. In practice, certain types of touchpoints can directly trigger uploads without being reflected in the proxy-mediated 'inspiration' pathway captured by $Y'$."**
> >
> > **Author response:** We appreciate this thoughtful concern. We agree that direct triggers such as creator incentives and reminder notifications are important in real-world CDP systems and should not be forced into the inspiration pathway. Our framework handles these effects by combining back-door and front-door strategies, and we clarify this separation more explicitly here.
> >
> > Concretely, variables such as exposure rewards, monetary incentives, and reminder notifications ("Your followers are waiting—post an update now!") are treated as observable covariates $X$ in our causal graph, rather than as part of the treatment $T$ (consumed touchpoints) or the mediator $M$. These factors can have a direct effect on $Y$ (uploads), but, in our CDP setting, they are not functions of which specific consumption events belong to $T$. In other words, they influence whether and when a creator uploads but are not themselves CDP "consume" events and thus are not part of the $T→M$ inspiration channel. In our implementation, $X$ includes both static user features and rich dynamic features, and we apply standard back-door adjustment (e.g., IPW with $X$, like causalMTA _Liu et al. (2025a)_) to remove bias from all observable confounders, including these direct triggers and the creator's observable propensity to post. Because back-door adjustment on $X$ is well established and not the conceptual novelty of our work, we did not emphasize it in the original manuscript, which may have contributed to the reviewer's concern.
> >
> > The front-door component, via the mediator $M$ approximated by $Y'$, is then used to handle the remaining unobserved confounding that cannot be exhausted by $X$. From the business definition of $Y'$ (shared music/templates/semantics between consumed and uploaded videos) and the high coverage discussed in Q1, the proxy-based mediator satisfies the qualitative front-door conditions along the inspiration path: (i) $T$ has a causal effect on $M$ (consumption influences inspiration), (ii) $M$ has a causal effect on $Y$ (inspiration influences uploads), (iii) there is no substantial unmeasured confounding between $T$ and $M$, and (iv) there is no substantial unmeasured confounding between $M$ and $Y$ after controlling for $T$, once observable triggers in $X$ have been adjusted for. Under this "back-door first, front-door second" strategy, direct triggers in $X$ are handled by back-door adjustment, and the front-door mechanism focuses on the residual unobserved system-level confounders.
> >
> > If, hypothetically, some of the reviewer's examples were not included in $X$ and were also not captured by $Y'$, then they would represent omitted confounders and could bias any causal estimate. In such a case, the front-door estimator can be more sensitive to violations than a pure back-door estimator, which is why in large-scale industrial practice we strongly prioritize modeling all observable triggers in $X$ before relying on the front-door component to handle what remains. However, in our deployed system, creator incentives and reminders are part of $X$ and do not affect $(T→M)$ or $(M→Y)$ directly; for example, whether a creator receives an incentive or reminder does not change whether previous consumption events share a particular music or template $(T→M)$, nor does it change whether the presence of such inspirational consumption leads to an upload $(M→Y)$. This means that, after adjusting for $X$, the inspiration pathway via $M/Y'$ remains a valid target for front-door-style adjustment.

---

> > > ### Comment · Reviewer_uog4 · 2025-11-28
> > >
> > > Thanks for the detailed and clear explanations from authors to my two questions. I have a better understanding of the idea of the motivation and method of this paper now.

---

### Official Review · Reviewer_JKZ9 · 2025-11-01

**Soundness:** 2
**Presentation:** 3
**Contribution:** 2
**Rating:** 4
**Confidence:** 3

**Summary:**

The paper proposes using front-door identification to handle latent confounders in recommendation systems and innovative adversarial proxy mediator design. This method reduces variance in uplift estimation by conditioning on the proxy. It is claimed that the method is designed to be scalable with real-world performances. The paper presented business results on DAU, daily active creators, and an upload AUC of 0.907 (40% increase on sota). The paper performs an analysis on the result, presenting a propensity-stratified grouped AUUC protocol using Shapley value sampling, and the algorithm is able to combine multiple sophisticated techniques with ablation studies.

**Strengths:**

- The paper describes a novel application of the front-door identification causal framework to handle latent confounders in recommender systems.
- The methods are experimented on a real-world online recommendation system, showing significant improvement.
-The paper combines multiple complex methods together and shows ablation on each factor.

**Weaknesses:**

- It is less clear in the writing how the proposed methods align, compare, or improve upon methods in prior art
- Compared with prior methods, other than Table 2, it could be useful to compare on more than one dataset or task, and perform experiments to show how the proposed methods work better and the specific reason why (e.g., due to front-door ID, adversarial mediator design, or scalability.
- Presentation: Figure 5 and Figure 6 are a bit hard to read
- it's a bit unclear whether the production or real-world data is made public for future work to compare to this paper

**Questions:**

- For comparison on other datasets and tasks, could more experiments be validated or provided to compare the proposed method against prior art?
- Could the presentation be clearer on the figures?
- Is the real-world data used made public for research comparisons?

---

> ### Author Response · Authors · 2025-11-27
> **Response to R1-1: "It is less clear in the writing how the proposed methods align, compare, or improve upon methods in prior art."**
>
> **Author response:** We appreciate this valuable feedback. We address the comparison with prior art from three perspectives:
>
> **1. Experiments in Original Submission**
>
> In the original manuscript, we conducted comprehensive comparisons against SOTA baselines (DeepMTA, CAMTA, CausalMTA) on our real-world CDP dataset using multiple evaluation metrics: (1) **AUC/gAUC** for discriminative performance, (2) **UAUC/gAUUC** for causal uplift ranking accuracy, and (3) **online A/B test metrics** including Author DAU Lift. Results demonstrate that ALM-MTA achieves significant improvements across all metrics. Notably, the online Author DAU improvement increased from approximately +0.1% (previous back-door adjustment scheme) to +0.7%, validating the practical business value.
>
> **2. Supplementary Experiments on Public Dataset**
>
> To further strengthen the generalizability of our findings, we conducted additional experiments on the public **Criteo Attribution Modeling Dataset**:
>
> _CPA means Cost Per Action, a lower Cost Per Action (CPA) under varying budget constraints indicates enhanced efficiency and a higher Return on Investment (ROI) for the advertising campaign._
>
> | Model | AUC | logloss |
> |-------|-----|---------|
> | CausalMTA | 0.9659±0.01 | 0.0517±0.003 |
> | ALM-MTA | 0.9729±0.01 | 0.0634±0.002 |
>
> | Model | Metric | 1/2 Budget | 1/4 Budget | 1/8 Budget | 1/16 Budget |
> |-------|--------|------------|------------|------------|-------------|
> | DeepMTA | CPA | 36.25 | 30.60 | 26.08 | 25.97 |
> | CausalMTA | CPA | 30.34 | 29.52 | 26.45 | 25.47 |
> | **ALM-MTA** | **CPA** | **27.32** | **24.85** | **22.71** | **22.03** |
>
> ALM-MTA achieves optimal CPA across all budget conditions, with more pronounced advantages under constrained budgets, demonstrating sensitivity to causal cost signals rather than spurious correlations.
>
> **3. Enhanced Ablation Analysis**
>
> We have restructured Section 5.5 to explicitly articulate the sources of performance gains:
>
> - **DML baseline** (AUC 0.6498, UAUC 0.50): Back-door adjustment alone cannot resolve unobserved system-level confounding.
> - **Adding mediator branch with $Y'$** (AUC 0.97, UAUC 0.90): High apparent performance but with severe loss oscillations (Fig. 5a), indicating outcome leakage.
> - **Adding adversarial discriminator** (AUC 0.86, UAUC 0.71): Smooth convergence (Fig. 5b), confirming effective suppression of $M \leftarrow W \rightarrow Y$ back-door leakage.
> - **Adding contrastive learning** (AUC 0.907, UAUC 0.82): Improved overlap in billion-level treatment space, reducing uplift estimation variance.
>
> This analysis clearly links each component to its function: **front-door identification** addresses unobserved confounding, **adversarial design** ensures valid mediator construction, and **contrastive adaptation** handles treatment sparsity.
>
> **Author action:** We have (1) incorporated the Criteo dataset experiments into **Appendix A.5**, and (2) revised Section 5.5 to explicitly connect the visual evidence in Figure 5 with the methodological mechanisms, providing a step-by-step explanation of how each component contributes to performance improvements.

---

> ### Author Response · Authors · 2025-11-27
> **Response to R1-2: "For comparison on other datasets and tasks, could more experiments be validated or provided to compare the proposed method against prior art?"**
>
> **Author response:** Thank you for this helpful suggestion. We strongly agree that extending our evaluation to additional datasets would significantly strengthen the solidity and generalizability of our results. Our current submission primarily focuses on a single large-scale CDP dataset, as it allows us to measure online causal impact - a key motivation we will make more explicit in the revised Section 5. However, we acknowledge that ALM-MTA is dataset-agnostic and should be validated against public benchmarks.
>
> Therefore, despite the page limitations of the initial submission, we have conducted supplementary experiments on the public **Criteo (Criteo Attribution Modeling for Bidding Dataset) dataset** within this response letter to compare our proposed method against other SOTA baselines. The specific performance comparison is detailed below. We have incorporated these additional results into the **Appendix A.5** to provide a more comprehensive evaluation.
>
> * Criteo (Criteo Attribution Modeling for Bidding Dataset) dataset represents a sample of 30 days of Criteo live traffic data. Each line corresponds to one impression (a banner) that was displayed to a user. For each banner we have detailed information about the context, if it was clicked, if it led to a conversion and if it led to a conversion that was attributed to Criteo or not. Data has been sub-sampled and anonymized so as not to disclose proprietary elements. The average touchpoint sequence length is 2.681103978733547, the longest touchpoint sequence length is 880, and the 90th percentile of the touchpoint sequence length is 6.0; the touchpoint percentage is 0.026866; the click-through rate (CTR) is 0.361158; the conversion rate is 0.0489552; and the average cost-per-click (CPO) is 0.196429.
> * Using cost (bucketing value, which makes modeling easier to converge and further reduces information leakage) as $Y'$, adversarial learning is used together with impressions to eliminate shortcuts from $T→Y$ and $Y→Y'$. The variables after adversarial learning are used as observations of $M$; the longest sequence length is set to 16.
> * AUC and logloss: To ensure stability, the model incorporates adversarial learning, resulting in a larger initial loss and thus a higher logloss after convergence compared to causalMTA. ALM-MTA models the transformation better (i.e., has a higher AUC). Furthermore, the confidence intervals of ALM-MTA are more stable, primarily due to further optimization of the stability component. To some extent, ALM-MTA exhibits better interpretability and robustness than other baseline models, yielding more stable causal conclusions.
>
> |           | AUC         | logloss      |
> |-----------|-------------|--------------|
> | causalMTA | 0.9659±0.01 | 0.0517±0.003 |
> | ALM-MTA   | 0.9729±0.01 | 0.0634±0.002 |
>
> * Because of the introduction of cpp/attribute (adversarial learning leakage prevention) as a front-door observation signal, ALM-MTA is more sensitive to cost/attribute and can achieve better CPA. For CVR, the improvement of ALM-MTA is limited at 1/2 budget, but it is ALM-MTA at 1/4, 1/8 and 1/16 budget, further illustrating that ALM-MTA is more sensitive to cost, especially when the budget is limited.
>
>     * **CPA**: Cost per action
>     * **Conv**:  Conversion Number
>     * **CVR**: Conversion rate
>     * **1/2, 1/4, 1/8, 1/16**: Budget of 1/2, 1/4, 1/8, 1/16
>
> | model | type | 1/2 | 1/4 | 1/8 | 1/16 |
> |------|----------|---------|---------|---------|----------|
> | DeepMTA | **CPA** | 36.25 | 30.60 | 26.08 | 25.97 |
> | | **Conv** | 1372 | 880 | 549 | 289 |
> | | **CVR** | 0.1194 | 0.1202 | 0.1236 | 0.1249 |
> | CAMTA | **CPA** | 32.61 | 29.73 | 26.05 | 26.25 |
> | | **Conv** | 1270 | 864 | 538 | 211 |
> | | **CVR** | 0.1127 | 0.1160 | 0.1191 | 0.1166 |
> | CausalMTA | **CPA** | 30.34 | 29.52 | 26.45 | 25.47 |
> | | **Conv** | 1441 | 976 | 548 | 255 |
> | | **CVR** | 0.1247 | 0.1265 | 0.1305 | 0.1283 |
> | ALM-MTA | **CPA** | 27.32 | 24.85 | 22.71 | 22.03 |
> | | **Conv** | 1503 | 1070 | 557 | 291 |
> | | **CVR** | 0.1248 | 0.1271 | 0.1320 | 0.1298 |

---

> > ### Author Response · Authors · 2025-11-27
> > **Response to R1-3: "Compare on more than one dataset or task... show how the proposed methods work better and why (front-door ID, adversarial design, scalability)."**
> >
> > **Author response:** Thank you for this insightful comment.
> >
> > * Regarding additional datasets: Please refer to our response to **Q1 (R1-2)**, where we have provided comparative experiments on the public **Criteo dataset** to demonstrate the generalization capability of our method.
> >
> > * Regarding the source of performance gains: Concretely, we start from a DML counterfactual baseline with user features, propensity correction and privileged inputs, which achieves AUC 0.6498 and UAUC 0.50. This baseline controls observable confounders but cannot address unobserved system-level confounding or the extremely large treatment space. Adding a mediator branch directly supervised by the proxy $Y'$ dramatically increases apparent performance (AUC 0.97, UAUC 0.90), confirming that $Y'$ carries strong signal; however, as shown in Fig. 5(a), this variant exhibits large loss oscillations and non-convergent uplift estimates, indicating severe outcome leakage and unstable causal learning. Introducing an adversarial discriminator on $Y$ to "clean" the mediator resolves this issue: AUC becomes 0.86 and UAUC 0.71, and the loss curve in Fig. 5(b) becomes smooth and monotone, evidencing a valid front-door-like $T→M→Y$ pathway that mitigates unobserved confounding. Finally, we attach a MoCo-style contrastive learner on $(T, Y')$ pairs. This module pulls together treatments that share similar mediator signals $Y'$ and pushes apart unrelated ones, which increases effective overlap in sparse treatment cells and yields a more structured representation of the billion-level treatment space. By reducing variance and improving generalization of the uplift estimator, this contrastive adaptation further improves upload prediction to AUC 0.907 and UAUC 0.82 and increases the AUC of strict-criterion and semantic attribution categories to 0.634 and 0.700 respectively (with UAUC 0.622 and 0.689).
> >
> > **Author action:** We have revised Section 5.5 to expand the ablation-study description and explicitly link the observed performance gains to front-door identification, adversarial mediator design, and contrastive adaptation in the large treatment space.

---

> > > ### Author Response · Authors · 2025-11-27
> > > **Response to R1-4: "Figure 5 and Figure 6 are a bit hard to read." Could the presentation be clearer on the figures?"**
> > >
> > > **Author response:** Thank you for pointing this out. We apologize for the lack of clarity in the initial figures. We understand that the visual presentation and the interpretability of the plots are crucial. Here we provide a detailed explanation of Figure 5 and Figure 6:
> > >
> > > * Figure 5 (Training Dynamics & Convergence):
> > > This figure illustrates the impact of adversarial learning and compares the training dynamics (Loss and AUC) across different models.
> > >
> > >     * Plot Configuration: The x-axis represents training steps, and the y-axis represents sampled loss or AUC values (smoothing window size = 4). To visualize sampling variance, we included confidence bands: solid lines denote the mean, while shaded areas indicate the confidence intervals ($\pm 1.96 \times$ standard error).
> > >
> > >     * Fig. 5(a) vs. Fig. 5(b) (Impact of Adversarial Learning): These subplots display the loss for modeling $Y$ via aggregated uplift. Fig. 5(a) depicts the scenario where the proxy $Y'$ is directly observed without adversarial constraints. The loss curve shows no clear convergence trend, indicating outcome leakage. In contrast, Fig.5(b) introduces adversarial learning alongside $Y'$, resulting in significantly improved convergence. This comparison confirms that adversarial learning effectively prevents information leakage from $Y'$.
> > >
> > >     * Fig. 5(c) & Fig. 5(d) (Comparison with Baselines): These plots track the Loss and AUC evolution of ALM-MTA versus baseline methods. Notably, ALM-MTA exhibits a higher initial loss, primarily due to the multiple objectives in our Multi-Task Learning (MTL) framework. However, as training progresses with more data, ALM-MTA demonstrates a distinct performance advantage over the baselines.
> > >
> > > * Figure 6 (Attribution Stability Analysis):
> > >
> > >     * Objective: This figure analyzes the distribution of uplift scores predicted by different models across different random seeds (using identical training data, steps, and inference sets). The goal is to evaluate interpretability stability: verifying that causal explanations (attribution order and values) do not fluctuate due to random parameter initialization.
> > >
> > >     * Significance: In production environments, stability is crucial to ensuring that identical cases do not receive contradictory attribution results purely due to model randomness. Therefore, higher similarity in uplift distributions across seeds indicates superior model stability.
> > >
> > >     * Result: Classical causal models (DML, DESCN, CausalMTA) exhibit noticeable divergence in distributions across seeds. In contrast, ALM-MTA, which incorporates specific stability-oriented designs, maintains highly consistent uplift distributions. This demonstrates that our method achieves superior reproducibility and robustness against random initialization.
> > >
> > >
> > > **Author action:** To address the readability concern and ensure the figures are clearly interpreted, we have implemented three key changes in the revised manuscript:
> > >
> > > * Expanded Captions: We have rewritten the captions for both figures to be self-explanatory.
> > >
> > > * Text Revision in Section 5.5: We have rewritten the "Ablation Studies and Mechanism Analysis" paragraph. The revised text now explicitly links the visual evidence in Figure 5 to the mechanism of our method, providing a step-by-step explanation of how Front-Door Identification and Adversarial Design contribute to the model's stability and performance.

---

> > > > ### Author Response · Authors · 2025-11-27
> > > > **Response to R1-5: "It's a bit unclear whether the production or real-world data is made public for future work to compare to this paper and Is the real-world data made public for future comparisons?"**
> > > >
> > > > **Author response:** Yes. Both the dataset and code will be made public upon acceptance.
> > > > Due to company data-protection policy, we are not allowed to release logs before acceptance. If the paper is accepted, we will:
> > > >
> > > > * Release a de-identified version of the CDP dataset and data preprocessing scripts.
> > > >
> > > > * Open-source the full codebase, including IPW estimation, the ALM-MTA architecture, and evaluation toolkit (AUC, gAUC, grouped AUUC, and stability plots).
> > > >
> > > > **Author action:** We have added this commitment and a more detailed reproducibility statement in the Reproducibility section.

---

### Author Response · Authors · 2025-11-30
**Response to Area Chair: Summary of Topic1 and Topic2**

Dear Area Chair,

To facilitate your review, we provide below a structured summary of all reviewer concerns and our corresponding responses. This document consolidates the key issues raised, the modifications we have made, and the interactions between reviewers and authors. We have categorized all concerns into four thematic areas based on their nature. For detailed elaborations and complete discussion threads, please refer to the comments under each specific question in the OpenReview system.

Best Regards,

All authors

### **1. Insufficient Comparative Analysis  with State-of-the-Art Methods.**

_R1-1. "It is less clear in the writing how the proposed methods align, compare, or improve upon methods in prior art."_

**We addressed comparison from three aspects: (1) original experiments showing improvements over SOTA baselines with online Author DAU lift from +0.1% to +0.7%; (2) supplementary Criteo experiments demonstrating optimal CPA across all budget conditions (added to Appendix A.5); (3) restructured Section 5.5 with enhanced ablation analysis explicitly linking each component (DML → mediator → adversarial → contrastive) to its functional contribution.**

_R1-2. "For comparison on other datasets and tasks, could more experiments be validated or provided to compare the proposed method against prior art?"_

**We conducted supplementary experiments on the public Criteo Attribution Modeling dataset, demonstrating ALM-MTA achieves higher AUC (0.9729 vs. 0.9659) and superior CPA performance particularly under constrained budgets (1/4, 1/8, 1/16). Results incorporated into Appendix A.5.**

_R1-3. "Compare on more than one dataset or task... show how the proposed methods work better and why (front-door ID, adversarial design, scalability)."_

**We revised Section 5.5 to explicitly connect performance gains to front-door identification, adversarial mediator design, and contrastive adaptation in large treatment space, with step-by-step ablation showing how each component addresses specific challenges (unobserved confounding, leakage prevention, treatment sparsity).**

_R4-1. "The authors need to further clarify their innovation. The authors claim to have solved the unobserved system-level confounding factors in the recommendation ecosystem and overcome a core limitation of previous attribution methods. However, in the field of debiased recommendation systems, several works have proposed solutions to potential confounding factors , including the application of the front-door criterion "_

**We clarified innovation relative to existing debiasing methods (Huang et al., Xu et al.) in three aspects: front-door identification with latent mediators, adversarial leakage prevention, and scalability to billion-level treatment spaces. Comparison added to Section 2 (Related Work) with refined contribution statement in Section 1.**

### **2. Suggestions for Model Clarification and Experimental Improvement.**

_R1-4. "Figure 5 and Figure 6 are a bit hard to read." Could the presentation be clearer on the figures?"_

**We rewrote Figure 5 and 6 captions to be self-explanatory and revised Section 5.5 text to explicitly link visual evidence to methodological mechanisms.**

_R3-2. "Make the causal graph explicit. Present a minimal DAG, list the front-door conditions, and pinpoint which one is relaxed and how the proxy + adversarial block recovers it."_

**We added minimal DAG (Appendix A.1 Fig.7) with explicit front-door conditions. Condition (ii) "no back-door from $M$ to $Y$" is identified as the one relaxed in practice, with explanation of how proxy + adversarial learning recovers it.**

_R3-3. "Add proxy-quality sensitivity. Vary the correlation between the proxy and the latent mediator, and show how performance (especially gAUUC and online KPIs) decays."_

**We conducted proxy-quality sensitivity analysis across four proxy configurations, showing monotonic relationship between coverage/precision and both gAUUC and online DAU. Table added to Appendix A.1.**

_R3-4. "Discuss non-additive / order effects. Even a short section on how to extend to interaction-aware aggregation (e.g. attention over touch sequences) would ease concerns about the additivity assumption."_

**We stated in Section 4.3 that production uses linear aggregation of context-dependent uplifts for scalability. Discussion of interaction-aware extensions added to Limitations section.**

_R3-5. "Relate gAUUC to business KPIs. Show which buckets / segments show the strongest monotonic relationship."_

**Provided segment-level analysis establishing the relationship between gAUUC improvements and downstream business KPIs, identifying customer segments exhibiting the strongest monotonic performance relationships. Please see the detailed elaboration under this question.**

---

> ### Author Response · Authors · 2025-11-30
> **Response to Area Chair: Summary of Topic3 and Topic4**
>
> ### **3. Validity of Causal Assumptions and Proxy Variable Design**
>
> _R2-1. "The use of the proxy variable $Y'$ as a surrogate for the mediator $M$ is insufficiently justified. The paper does not provide rigorous theoretical or empirical evidence establishing the validity of $Y'$ and it remains unclear whether $Y'$ meets the necessary proxy conditions for front-door identification. More importantly, the paper does not prove that $Y'$ does not introduce a new path to the outcome $Y$, which could bias the causal effect estimation."_
>
> **We provided both empirical and theoretical justification for proxy $Y'$ validity: business-grounded definition (same music/template, semantic similarity) with ~60% coverage, and formal proxy conditions following Miao et al. (2018) and Tchetgen et al. (2024). Detailed discussion added to Appendix A.1 with pointer in Figure 2 caption.**
>
> _R2-2. "The applicability of the front-door criterion is assumed rather than validated. The key identification assumption—that all causal influence of touchpoints on uploads must operate exclusively through the latent mediator $M$ appears overly strong and may not hold in real-world CDP environments..."_
>
> **We clarified that direct triggers (creator incentives, reminder notifications) are treated as observable covariates $X$ with standard back-door adjustment (IPW), while front-door mechanism via $M$ handles residual unobserved confounding. This "back-door first, front-door second" strategy ensures direct triggers are addressed by back-door adjustment on $X$, and front-door focuses on system-level confounders that cannot be exhausted by $X$. Please see the detailed elaboration under this question.**
>
> _R4-2. "How do the authors distinguish between covariates $X$ and potential confounders $W$ in practice, and how do they ensure that the model does not regard $X$ as a confounder?..."_
>
> **We clarified that both $X$ and $W$ are confounders: $X$ represents observable confounders (user attributes, creative incentives) addressed via back-door criterion; $W$ represents unobservable confounders (cold start, distribution strategies) addressed via front-door criterion. Business meanings of each variable added to Figure 2 in Section 3.**
>
> _R4-3. "The authors regard the observed results of $Y$ as $Y'$ and use it as a proxy variable for $M$ to guide the generation of $M$. In my opinion, $Y'$ is almost equivalent to $Y$, and there is a risk of data leakage..."_
>
> **We clarified that $Y'$ is not identical to $Y$: $Y$ is binary upload outcome while $Y'$ is path-level inspiration signal (covers ~60% of uploads). Training dynamics in Figure 5(a)/(b) demonstrate adversarial learning effectively suppresses leakage. Please see the detailed elaboration under this question.**
>
> _R4-4-reply. "...I would like to know how you ensure the reliability of $M$. In other words, how do you guarantee that $M$ meets the requirements of the front-door adjustment?"_
>
> **We provided formal justification that $M$ satisfies front-door conditions through business logic and adversarial learning ensuring exclusion condition. Detailed verification added to Appendix A.1.**
>
> ### **4. Concerns Regarding Model Complexity and Reproducibility.**
>
> _R1-5. "It's a bit unclear whether the production or real-world data is made public for future work to compare to this paper and Is the real-world data made public for future comparisons?"_
>
> **Acknowledged proprietary constraints preventing public release of production data; released implementation code along with comprehensive replication guidelines for the Criteo benchmark dataset in supplementary materials to facilitate reproducibility.**
>
> _R3-1. "Complexity and reproducibility: The training pipeline optimizes multiple objectives at once (main prediction, proxy branch, adversarial loss, IPW-related parts, contrastive loss). This makes the system heavy and hard to reproduce for teams without mature ML/causal infra. Because of this complexity, the method is somewhat black-box: it is not straightforward to tell which part of unobserved bias was actually removed."_
>
> **We provided detailed training recipe in Appendix A.3: loss weight ratios (main : DML : adversarial : regularization : contrastive ≈ 1:0.6:4:0.1:0.2), warm-up order, and PCGrad-style gradient routing. Ablation analysis explicitly connects each component to the bias type it addresses.**
>
> _R4-4. "In Section 4, the authors did not describe the specific implementation details, but only provided a formal description. For example, the authors introduce adversarial learning but don't provide a detailed implementation. Similarly, the paper doesn't explain how the mediating variable $M$ is generated. This raises concerns about whether ALM-ATA can be replicated."_
>
> **Provided detailed implementation specifications including adversarial training procedures, mediator generation mechanisms, and hyperparameter configurations. Complete pseudocode and training protocols added in Appendix A.3.**

---

> ### Author Response · Authors · 2025-11-30
> **Response to Area Chair: Note on Reviewer-Author Interactions**
>
> **Reviewer 2 (uog4):**
>
> *  **Initial Concerns:** Questioned front-door criterion applicability and proxy validity, noting the assumption appears overly strong for real-world CDP environments.
>
> * **Author Response:** Clarified the “back-door first, front-door second” strategy, explaining that direct triggers (incentives, notifications) are handled via back-door adjustment on $X$, while front-door mechanism addresses residual unobserved confounding.
>
> * **R2 Follow-up:** _"Thanks for the detailed and clear explanations from authors to my two questions. I have a better understanding of the idea of the motivation and method of this paper now."_
>
> **Reviewer 4 (eh6j):**
>
> * **Initial Concerns:** Raised questions on implementation details, mediator generation, $Y'/Y$ distinction, and data leakage risk.
>
> * **Author Response:** Provided clarifications on business-grounded proxy construction, $Y'$ as path-level signal distinct from $Y$, and adversarial leakage prevention with training dynamics evidence (Figure 5a/b).
>
> * **R4 Follow-up:** _"Thank you for the detailed responses, which have addressed some of my questions and concerns. I now understand how the mediator variable $M$ is generated. However, I would like to know how you ensure the reliability of $M$. In other words, how do you guarantee that $M$ meets the requirements of the front-door adjustment? If the authors can clarify this point, I would be willing to raise my score."_
>
> * **Author Further Response:** Provided formal justification that $M$ satisfies front-door conditions through business logic and adversarial learning, with detailed verification in Appendix A.1.

---

### Meta-Review · Area_Chair_528C · 2026-01-19

**Summary:**

The paper presents a new method for multi-touch attribution in recommender systems under unobserved confounding. The method uses a distilled proxy outcome obtained via adversarial learning, with a contrastive learning contrastive learning block to handle large treatment spaces. The method is evaluated on a real-world recommendation system with 400 million daily active users and 30 billion samples.

The main weaknesses identified by the reviewers are:

W1. Insufficient positioning with respect to (and comparisons against) stage of the art methods.

W2. Evaluation performed on a single dataset, which appears to be private.

W3. Insufficient justification and theoretical guarantees for the proxy variable.

W4. Missing implementation details, coupled with the complexity of the method make the paper difficult to understand and reproduce.

**Reviewer Concerns:**

In their response, the authors have, for the most part, addressed the concerns to a degree which I find satisfactory.

W1. The authors provided further details on differentiation to proper work in their responses to reviewers eh6j and Reviewer JKZ9. I find these clarifications to be useful. It is possible that there are other techniques that the authors might want to compare against, but reviewer JKZ9 did not mention any references.

W2. An experiment on the Criteo datasets was added, showing that the method outperforms the state of the art. This addresses the issues concerning replicability and should definitely be included in the final version of the paper.

W3. The authors have provided further details in the response to reviewer R2-1, who acknowledged that the response improved their understanding of the motivation and idea, indicating the response was satisfactory.

W4. Additional details were provided in the answers to reviewers R3-1 and eh6j. Reviewer eh6j confirmed that the answer facilitates their understanding, and asked one final question about the reliability of the mediator variable, indicating that they would raise the score if this point is clarified. The authors have followed up explaining why  "there is no unblocked backdoor path from T to M; all backdoor paths from M to Y are blocked by T; They also explain that M completely mediating the effect of T on Y is "grounded in a key business assumption". I find these clarifications comprehensive and clear.


Other weaknesses were either minor or explained adequately by the authors.


Overall, this paper provides a valuable contributions to large scale online recommender systems which work with complex treatments. The real-life experiment is impressive, albeit there are clear issues with data sharing. However, the authors have provided experiments on a public dataset, demonstrating the performance of their method and alleviating those concerns. The authors also clarified how their paper differs from related work and provided further details about the method.

**Reviewer Scores:**

I do not know how the reviewers would have changed their scores.

---

### Decision · Program_Chairs · 2026-01-26

Accept (Poster)